# Influence of convection on the upper tropospheric $O_3$ and $NO_x$ budget in southeastern China

Xin Zhang[1,2], Yan Yin[1,2], Ronald van der A[2,3], Henk Eskes[3], Jos van Geffen[3], Yunyao Li[4], Xiang Kuang[1,2], Jeff L. Lapierre[5], Kui Chen[1,2], Zhongxiu Zhen[1,2], Jianlin Hu[1,2], Chuan He[1,2], Jinghua Chen[1,2], Rulin Shi[6], Jun Zhang[7], Xingrong Ye[7], and Hao Chen[7]

[1] Collaborative Innovation Center on Forecast and Evaluation of Meteorological Disasters/Key Laboratory for Aerosol-Cloud-Precipitation of China Meteorological Administration, Nanjing University of Information Science and Technology (NUIST), Nanjing 210044, China
[2] Department of Atmospheric Physics, Nanjing University of Information Science and Technology (NUIST), Nanjing 210044, China
[3] Royal Netherlands Meteorological Institute (KNMI), Department of Satellite Observations, De Bilt, the Netherlands
[4] Department of Atmospheric, Oceanic & Earth Sciences, George Mason University, Fairfax, VA, USA
[5] Earth Networks, Germantown, Maryland, USA
[6] Inner Mongolia Lightning Warning and Protection Center, Hohhot, 010051, China
[7] Nanjing National Reference Climatological Station, Nanjing 210044, China

**Correspondence:** Yan Yin (yinyan@nuist.edu.cn)

**Abstract.** Thunderstorms can significantly influence the air composition via strong updraft and lightning nitrogen oxides ($LNO_x$). In this study, the ozonesondes and TROPOMI nitrogen dioxide ($NO_2$) observations for two cases are combined with model to investigate the effects of typical strong convection on vertical redistribution of air pollutants in Nanjing, southeastern China. The ozonesonde observations show higher $O_3$ and water vapor mixing ratios in the upper troposphere (UT) after

convection, indicating the strong updraft transporting lower-level airmass into the UT, and the possible downward $O_3$-rich air near the top of UT over the convective period. During the whole convection life cycle, the UT $O_3$ production is driven by the chemistry (5–10 times the magnitude of dynamic contribution) and reduced by the $LNO_x$ (−40 %). Sensitivity tests demonstrate that neglecting $LNO_x$ in standard TROPOMI $NO_2$ products causes overestimated air mass factors over fresh lightning regions and the opposite for outflow and aged lightning areas. Therefore, a new high-resolution retrieval algorithm is applied to

estimate the $LNO_x$ production efficiency. Our work shows the demand for high-resolution modeling and satellite observations on $LNO_x$ emissions of both active and dissipated convection, especially small-scale storms.

## 1 Introduction

Convection can transport the surface pollutants and moisture from the planetary boundary layer to the upper troposphere (UT) in a short time, where the gaseous pollutants have a longer lifetime due to the slower reaction rates, except for photolysis, in

the colder environment (Dickerson et al., 1987). As trace gases remain for more than one week in the UT, they are distributed by the upper level winds around the globe (Ridley et al., 2004). Meanwhile, the vertical profiles of trace gases are reshaped by the updraft and downdraft on timescales of hours (Huntrieser et al., 2016; Barth et al., 2019). On a global scale, the chemical

reactions of transported ozone ($O_3$) and its precursors can increase the amount of UT $O_3$ (Lawrence et al., 2003; Murray, 2016).

Nitrogen oxides ($NO_x$), volatile organic compounds (VOC), and hydrogen oxide radicals ($HO_x$) are the three main $O_3$ precursors, which can be uplifted or produced by thunderstorms (Bozem et al., 2017). Lightning produced nitrogen oxides ($LNO_x$) is the dominant natural source of UT $NO_x$, contributing as much as 35–45 % of global free-tropospheric ozone (Allen et al., 2010; Liaskos et al., 2015). Several lightning parameterizations have been developed for chemistry transport and climate models to evaluate the relationship between $LNO_x$ and other trace gases (Murray et al., 2012; Gordillo-Vázquez et al.,

2019; Luhar et al., 2021). Therefore, the precise estimation of $LNO_x$ is crucial for the global $O_3$ trend and feedback between lightning and climate change (Finney et al., 2016; Chen et al., 2021). Globally, the $LNO_x$ is estimated as 2–8 Tg N $yr^{-1}$, which is substantially less well quantified than the flash rate (Schumann and Huntrieser, 2007).

The large uncertainty of $LNO_x$ estimation might be reduced by using cloud-resolving chemistry models in combination with satellite and aircraft observations. It is beneficial to take advantage of sonde and satellite observations for exploring the

convection effects, especially for the estimation of the $LNO_x$ production efficiency (PE). Recently, many studies have used different satellites to quantify the $LNO_x$ PEs (Beirle et al., 2009; Pickering et al., 2016; Zhang et al., 2020). Two main methods have been proposed to distinguish $LNO_x$ from the $NO_2$ background pollution: 1) subtracting the weighted temporal average $NO_2$ of areas with few flashes before the satellite passing time (Pickering et al., 2016; Bucsela et al., 2019; Allen et al., 2019; Lapierre et al., 2020) and 2) directly using customized lightning air mass factors (AMFs) for each convection event (Beirle

et al., 2009; Zhang et al., 2020). Recently, Allen et al. (2021) proves the potential of deriving $LNO_x$ PEs by the geostationary lightning instruments (e.g., Lightning Mapping Imager (LMI; Yang et al., 2017), Geostationary Lightning Mapper (GLM; Rudlosky et al., 2019)), and $NO_2$ observations such as Tropospheric Emissions: Monitoring of Pollution (TEMPO; Chance et al., 2019).

Furthermore, aircraft observations and chemical models indicate that the transport from the stratosphere to the troposphere

can also increase the UT $O_3$ besides the chemical production from $LNO_x$ (Pan et al., 2014). As revealed in the Deep Convective Clouds and Chemistry 2012 Studies (DC3) and mesoscale convective system simulations, the compensation of subsidence and differential advection beneath the convective core can lead to the anvil wrapping effects (Huntrieser et al., 2016; Phoenix et al., 2020). The different mechanisms of stratosphere–troposphere exchange and the effects on the tropospheric chemistry have been discussed in Holton et al. (1995) and Stohl (2003).

At present, most aircraft observations and model simulations of convection effects focus on the tropics or the United States (Vaughan et al., 2008; Barth et al., 2019). Little is known about the role of convection in southeastern China (Murray, 2016; Guo et al., 2017), where thunderstorm and lightning have increased significantly by urbanization during recent decades (Yang and Li, 2014; Pérez-Invernón et al., 2021). This is likely due to the increasing aerosol concentration, which can invigorate storms in a moist and convectively unstable environment (Koren et al., 2008; Rosenfeld et al., 2008; Tao et al., 2012). In this

study, we combine ground observations and model simulations to investigate the origin of higher UT $O_3$ and water vapor mixing ratio (Qv) after convection, and we try to distinguish the contributions of physical processes, chemical reactions and $LNO_x$. For the first time, the TROPOspheric Monitoring Instrument (TROPOMI; Veefkind et al., 2012) $NO_2$ observations

are used to identify LNO$_x$ PEs in southeastern China. Section 2 describes the used datasets with a brief introduction of the cloud-resolved chemistry model and the LNO$_x$ retrieval method. Section 3 evaluates the model simulations and the physical and chemical effects of convection are analyzed in Sect. 4 and 5. We apply new a priori NO$_2$ profiles into the retrieval algorithm to explore the sensitivity of AMFs to LNO$_x$ in Sect. 6. Conclusions are summarized in Sect. 7.

## 2 Datasets

### 2.1 Ozonesonde data

Five ozonesondes were launched from the Nanjing National Reference Climatological Station (31.93° N, 118.90° E) on 25 July 2019 and 01 September 2020, both days experienced strong convection. Both pre-convection and during-convection/post-convection campaigns were designed to investigate the convection effects. The convection and ozonesonde trajectories are illustrated in Fig. 1a and b.

Three Institute of Atmospheric Physics (IAP) ozonesondes had been launched near the airmass convection that developed on 25 July 2019. The IAP ozonesonde uses an electrochemical concentration cell (ECC). The complete parameters and performance are described in Zhang et al. (2014). Its average bias is less than 0.3 mPa from the surface up to 2.5 km, close to zero below 9 km, and less than 0.5 mPa between 9 km and 18 km. The first IAP ozonesonde was launched at 05:35 UTC on a sunny day (23 July) and the other two were at 05:10 UTC (pre-convection) and 06:35 UTC (post-convection) on 25 July. Because of water leakage, the pre-convection one lost signal just a few seconds after the release, and instead the ozonesonde launched on 23 July is chosen. Although the time interval is two days, the largest relative difference of forecast O$_3$ profiles above 10 km is usually smaller than 25 % (Fig. S2), therefore the daily variation cannot explain the observed difference of more than 65 %.

The two Vaisala ECC ozonesondes were launched successfully at 23:45 UTC 31 August (pre-convection) and 06:10 UTC 01 September (during-convection), respectively, following the standard manual to ensure the precision is better than 5 % and the accuracy is within ±(5–10) % below 30 km (Smit et al., 2007). The captured squall line was developing from the convergence of cold air and typhoon Maysak's outer region circulation. Note that the during-convection ozonesonde entered directly into the cloud, providing a unique opportunity of exploring the ozone affected by the convective clouds.

### 2.2 Lightning data

Three lightning datasets were used in this study: the China National Lightning Detection Network (CNLDN; Yang et al., 2015), the Earth Networks Total Lightning Network (ENTLN; Marchand et al., 2019), and the World Wide Lightning Location Network (WWLLN; Rodger et al., 2006). The detection efficiency (DE) of cloud-to-ground (CG) flashes is about 90 % for the CNLDN data in Jiangsu province (Li et al., 2017a) while ENTLN and WWLLN detect both intra-cloud (IC) and CG flashes with specific detection frequency (1 Hz–12 MHz for ENTLN and 3–30 kHz for WWLLN). In the ENTLN data, groups of pulses are classified as a flash if they are within 700 ms and 10 km. Both strokes and lightning flashes composed of one or more strokes are included in the preprocessed data obtained from the ENTLN. The detailed processing algorithm of the

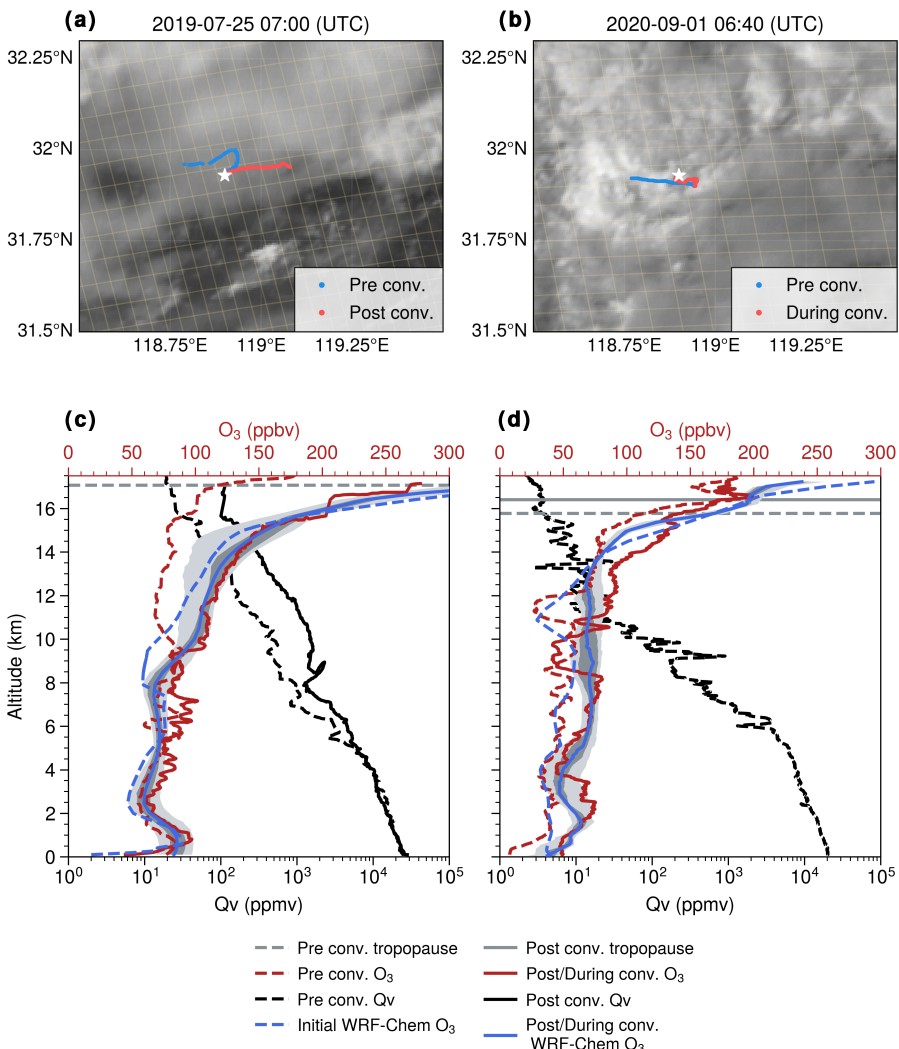

**Figure 1.** (a, b) The convection detected by the FY-4A Advanced Geostationary Radiation Imager (AGRI) visible channel (0.65 $\mu$m) field at the time when the post-convection/during-convection ozonesondes reached around 10 km. The pre-convection ozonesonde trajectories are colored blue while others are in red. The white star symbol stands for the observation station and the thin yellow lines are the TROPOMI swath pixels. (c) and (d) are the observed $O_3$ (red) and $Qv$ (black) profiles in the pre-convection (dashed) and post-convection/during-convection (solid) periods. The initial (dashed) and simulated post-convection or during-convection (solid) $O_3$ profiles are in blue. The dark gray shading is the 50 % confidence interval while the light one is the 90 % confidence interval. The gray lines are the lapse rate tropopauses.

WWLLN is given by Rodger et al. (2004). The WWLLN strokes and pulses are combined with ENTLN into one dataset (ENGLN) within 10 km and 0.7 s as mentioned in Virts and Goodman (2020). To increase the lightning data coverage in our study, the CG flashes of ENGLN and CNLDN datasets are combined using spatial and temporal clustering criteria of 10 km

and 0.5 s (Zhao et al., 2020). Merging these three datasets should provide a sufficiently high CG flash detection efficiency for this analysis. Because the IC DE of all these lightning data is low in China, we conservatively use the merged CG data with a constant IC/CG ratio of 3:1 based on Wu et al. (2016) and Bandholnopparat et al. (2020). IC data will become more accurate

if more Chinese total lightning networks, such as Beijing Lightning Network (BLNET; Srivastava et al., 2017), are available to be compared with lightning imaging sensors (Rudlosky and Shea, 2013; Poelman and Schulz, 2020).

## 2.3 TROPOMI data

On 13 October 2017, the TROPOMI on Sentinel-5 Precursor satellite was launched successfully (Veefkind et al., 2012). For the current study, we used the Royal Netherlands Meteorological Institute (KNMI) standard product v2.1-test as input to our

$LNO_x$ retrieval algorithm. A spike removal is included in the product to better deal with the detector saturation and blooming effects, which enables more valid data over the bright clouds generated by convection (Ludewig et al., 2020; van Geffen et al., 2022). Each pixel includes a slant column density quality flag (no2_scd_flag=0, see Appendix A) that can be used to get the data without known retrieval errors (Allen et al., 2021; van Geffen et al., 2022).

The official $NO_2$ columns are retrieved using the near-ultraviolet and visible (UV-VIS, 405–465 nm) spectrometer backscat-

100 tered solar radiation measurements on the TROPOMI (van Geffen et al., 2015). The retrieval consists of three main procedures for each measured Level-1b spectrum:

1) Total $NO_2$ slant column density (SCD) is determined by the DOAS method.

2) The stratospheric and tropospheric SCDs are separated by data assimilation of slant columns in the Tracer Model, version 5, tailored for the application of satellite retrievals (TM5-MP; Williams et al., 2017).

3) The stratospheric and tropospheric $NO_2$ vertical column density (VCD) are obtained via the air mass factor (AMF) look-up tables (Lorente et al., 2017).

We replaced the tropospheric AMF ($AMF_{trop}$) with a new AMF called AMF for $LNO_x$ ($AMF_{LNO_x}$) to derive the tropospheric $LNO_x$ vertical column density ($VCD_{LNO_x}$). The concept of $AMF_{LNO_x}$ inherits from the $AMF_{trop}$ derived by a function of several parameters (solar zenith angle, viewing zenith angle, relative azimuth angle, surface albedo, surface pressure, cloud fraction,

cloud height, and a priori trace gas profile). Briefly, the numerator is the modeled tropospheric $NO_2$ slant column density ($SCD_{tropNO_2}$) and the denominator is the modeled VCD ($VCD_{NO_2}$ or $VCD_{LNO_x}$). In detail, these two AMFs can be calculated as:

$$AMF_{\text{Trop}} = \frac{(1-f_r)\int_{p_{surf}}^{p_{tp}} w_{clear}(p)NO_2(p)\,dp + f_r\int_{p_{cloud}}^{p_{tp}} w_{cloudy}(p)NO_2(p)\,dp}{\int_{p_{surf}}^{p_{tp}} NO_2(p)\,dp} \tag{1}$$

$$AMF_{\text{LNO}_x} = \frac{(1-f_r)\int_{p_{surf}}^{p_{tp}} w_{clear}(p)NO_2(p)\,dp + f_r\int_{p_{cloud}}^{p_{tp}} w_{cloudy}(p)NO_2(p)\,dp}{\int_{p_{surf}}^{p_{tp}} LNO_x(p)\,dp} \tag{2}$$

where $p_{surf}$ is the surface pressure, $p_{tp}$ is the tropopause pressure, $p_{cloud}$ is the cloud optical pressure, $f_r$ is the cloud radiance fraction in the $NO_2$ window, $w_{clear}$ and $w_{cloudy}$ are respectively the pressure-dependent scattering weights from the

lookup table (Lorente et al., 2017) for clear and cloudy parts, and $NO_2(p)$ is the NO$_2$ vertical profile simulated by WRF-Chem. Besides, $LNO_x(p)$ is the LNO$_x$ vertical profile calculated by the difference of vertical profile between WRF-Chem simulations with and without lightning. All other parameters in the KNMI v2.1-test product, including the total SCD, stratospheric SCD, total VCD, stratospheric VCD, surface albedo, and scattering weights, remain unchanged.

In comparison with this study, Pickering et al. (2016), Allen et al. (2019), Bucsela et al. (2019), and Allen et al. (2021) defined another AMF$_{LNO_x}$ to convert SCD$_{tropNO_2}$ to the tropospheric NO$_x$ vertical column density (VCD$_{NO_x}$). Then, their VCD$_{LNO_x}$ can be calculated as the VCD$_{NO_x}$ subtracted by a tropospheric NO$_x$ background. Because our AMF$_{LNO_x}$ converts the SCD$_{tropNO_2}$ to VCD$_{LNO_x}$ directly, the additional estimation of background NO$_2$ is not needed for calculating LNO$_x$ PE in Sect. 6.2.

## 2.4   Model simulations

This study uses Weather Research and Forecasting model with chemistry (WRF-Chem) version 4.1.4. The initial and boundary conditions of meteorological parameters are provided by the hourly ECMWF atmospheric reanalysis (ERA5) data (Hersbach et al., 2020). Simulations are performed with one-way nesting with 75 vertical levels and a 50 hPa model top. The domain settings are illustrated in Fig. S1. The microphysical processes are computed with the WRF Single-Moment 6-class scheme (WSM6; Hong and Lim, 2006), while the shortwave and longwave radiation is calculated by the Rapid Radiative Transfer Model for GCMs scheme (RRTMG; Iacono et al., 2008). The land surface processes are simulated by the Noah scheme (Koren et al., 1999). However, we use different planetary boundary layer (PBL) parameterizations to simulate the convection. Specifically, the 2019 case uses the Yonsei University scheme (YSU; Hong et al., 2006), while the Quasi-Normal Scale Elimination (QNSE; Sukoriansky et al., 2005) is applied to the 2020 case.

The chemical initial and boundary conditions are defined using the output from the Whole Atmosphere Community Climate Model (WACCM, https://www.acom.ucar.edu/waccm/, last access: April 12, 2022). The initial O$_3$ profile of the 2020 case is replaced by the O$_3$ profile from the ozonesonde. Anthropogenic emissions are driven by the 2016 Multi-resolution Emission Inventory for China (MEIC) version 1.3 (http://www.meicmodel.org/, last access: April 12, 2022). The Model of Emissions of Gases and Aerosol from Nature (MEGAN; Guenther et al., 2006) is used for biogenic emissions. The chemical mechanism is the Model for Ozone and Related chemical Tracers (MOZART) gas phase chemistry and Goddard Chemistry Aerosol Radiation and Transport aerosols (GOCART) for aerosols (Pfister et al., 2011). The photolysis rates are adjusted by the presence of aerosols and clouds using the new TUV photolysis option with the scaled cloud optical depth (cloud_fraction$^{1.5}$). Note that the bimodal profile modified from the standard Ott et al. (2010) profile is employed as the vertical distribution of LNO in WRF-Chem (Laughner and Cohen, 2017), while the LNO$_x$ parameterization is activated as 500 mol NO per flash (Zhu et al., 2019). The resulting lightning nitrogen monoxide (LNO) and lightning nitrogen dioxide (LNO$_2$) profiles are defined as the difference of vertical profiles between simulations with and without lightning.

To simulate the convection and LNO$_x$ realistically, a lightning data assimilation (LDA) technique is applied to WRF-Chem. The details of the LDA technique can be found in Fierro et al. (2012) and Li et al. (2017b) and are briefly illustrated here. The water vapor mass mixing ratio is increased at constant temperature layers in columns where flashes occur:

$$Q_v = AQ_{\text{sat}} + BQ_{\text{sat}} \tanh(CX) \left[1 - \tanh\left(DQ_g^\alpha\right)\right] \tag{3}$$

where Q$_{\text{sat}}$ is the water vapor saturation mixing ratio (g kg$^{-1}$), Q$_g$ is the graupel mixing ratio (g kg$^{-1}$) and X is the flash rate. In our simulations, the layer between 263.15 K and 290.15 K is chosen to let the convection root in the PBL quickly as the Q$_v$ in the lower troposphere is the deeper layer (Marchand and Fuelberg, 2014; Finney et al., 2016; Li et al., 2017b). Parameter settings follow Li et al. (2017b): A = 0.94, B = 0.2, C = 0.001 and D = 0.25 and $\alpha$ = 2.2. The resampled total lightning flashes data are read through the Auxiliary Input Stream of WRF every 10 minutes. For example, if the beginning time of LDA is 05:00 UTC with a time step of 10 min, all flashes in a specific grid between 05:00 and 05:10 UTC are summed as the contribution during this period. At the next time step, the flashes are classified as the next new group. Therefore, the flash count is the flash rate density (units: flashes 10 min$^{-1}$ dx km$^{-1}$ dy km$^{-1}$), where dx and dy are the resolutions of model grids in the x and y direction, respectively. This leads to flash rate densities that are the same through all nested domains as done in Fierro et al. (2012) and Li et al. (2017b), and these are used as the flash counts in the WRF-Chem directly following the method employed in CMAQ (Kang et al., 2019a, b, 2020).

## 3   Model evaluation

Compared with the radar observation, the simulated initiations of convection are ahead by 60 minutes and 30 minutes for the 25 July 2019 and 01 September 2020 cases, respectively (Fig. 2 and Fig. 3). The lightning assimilations are applied with the same time step forwards. For comparisons below, we choose the matched stages instead of the same time.

The airmass convection on 25 July 2019 was initialized as isolated cells. The WRF-Chem reproduces the position and intensity of isolated convections at the initial stage (Fig. 2a and 2d). At 05:40 UTC, the cells were presented with the northeast-southwest orientation and the column maximum radar reflectivity (CRF) reached 60 dBZ (Fig. 2b), which is stronger than the simulated convection (CRF = 55 dBZ, Fig. 2e). The vertical cross section of simulated radar reflectivity across the core of cells is compared with the observation (Fig. S3). Although there were much missing data caused by the long distance between convection and radar, the horizontal and vertical structures of isolated cells are roughly shown without artificial interpolation. While the simulated 45 dBZ contour extends to 12 km, the observed one only reaches 10 km because of reduced data quality above 10 km.

The squall line on 1 September, 2020 was born in the north, strengthened, and moved towards the observation site (Fig. 3). The strongest convective stage with a CRF of 60 dBZ was at 05:50 UTC which is the TROPOMI overpass time (Fig. 3b and 3e). Although the highest level reached was lower than that of 2019 case, the reflectivity in the lower troposphere (2–8 km) was larger and broader (Fig. S4). Note that the simulated dissipated cells deviate from the radar observation and this leads to the region for the ozonesonde comparison moving to the west of station (Fig. 3c and 3f).

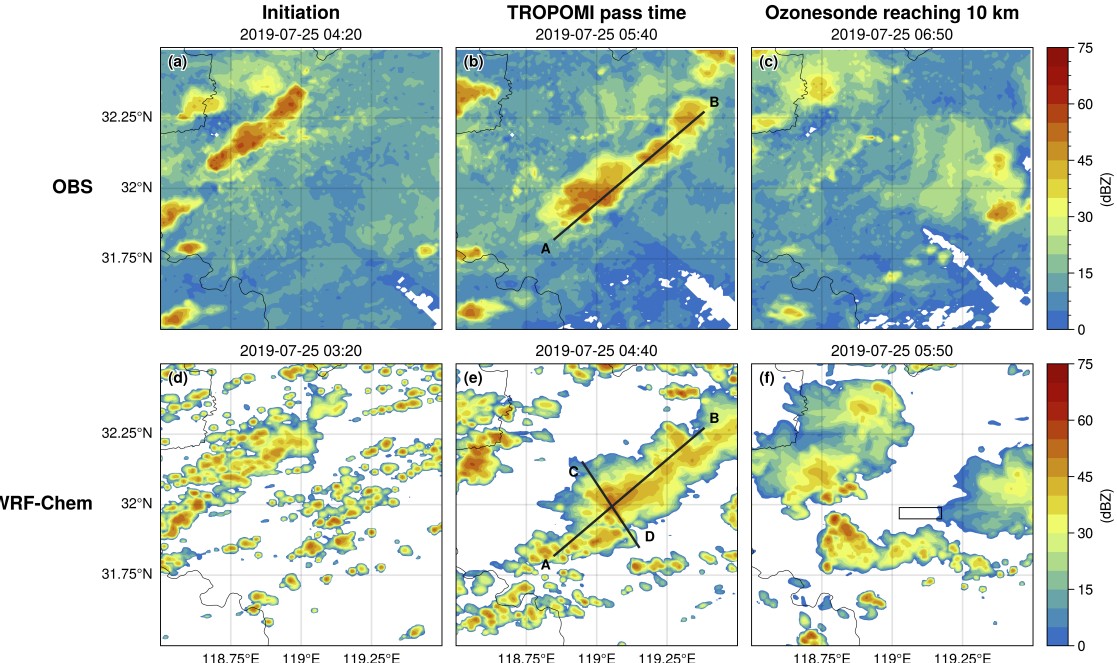

**Figure 2.** Observed radar composite reflectivity at (a) 04:20 UTC, (b) 05:40 UTC, and (c) 06:50 UTC. (d–e) WRF-Chem simulated composite reflectivity one hour before the radar observation times. The AB solid lines in (b) and (e) are cross section lines for Fig. S3. The CD solid line in (e) is the cross section line for Fig. 4b. The black rectangle is the region for the comparison with ozonesonde.

The measured $O_3$ and Qv profiles at different convection stages are shown in Fig. 1c and d. Generally, the observed UT $O_3$ and Qv are higher with convection, while the largest enhancements are between 10 km and 16 km. However, the 2020 case showed a larger increase in the lower troposphere (2–8 km, LT). Additionally, a two-valley shape of $O_3$ profile exists in both cases but at different levels: 2/8 km for the 2019 airmass and 4/10 km for the 2020 squall line. Although the WRF-Chem model tends to underestimate the $O_3$ concentration in the LT and UT for the 2019 and 2020 cases, respectively, it can reproduce the detailed $O_3$ structures and provide the opportunity to analyze the mechanisms of convection.

Three possible sources can explain the enhancements of $O_3$ in the UT: convective transport, chemical production, and $O_3$ directly produced by lightning. Only the first two factors are discussed in detail in Sect. 4, as lightning $O_3$ is beyond the scope of this study and still uncertain as shown by limited observations and model simulations (Morris et al., 2010; Ripoll et al., 2014).

## 4  Convection impacts

To evaluate the higher UT $O_3$ concentration after the convection, the mean vertical profiles of $O_3$ in the atmospheric regions passed by the ozonesondes are illustrated at three stages of convection: initiation, development, and dissipation (Fig. 4a and d).

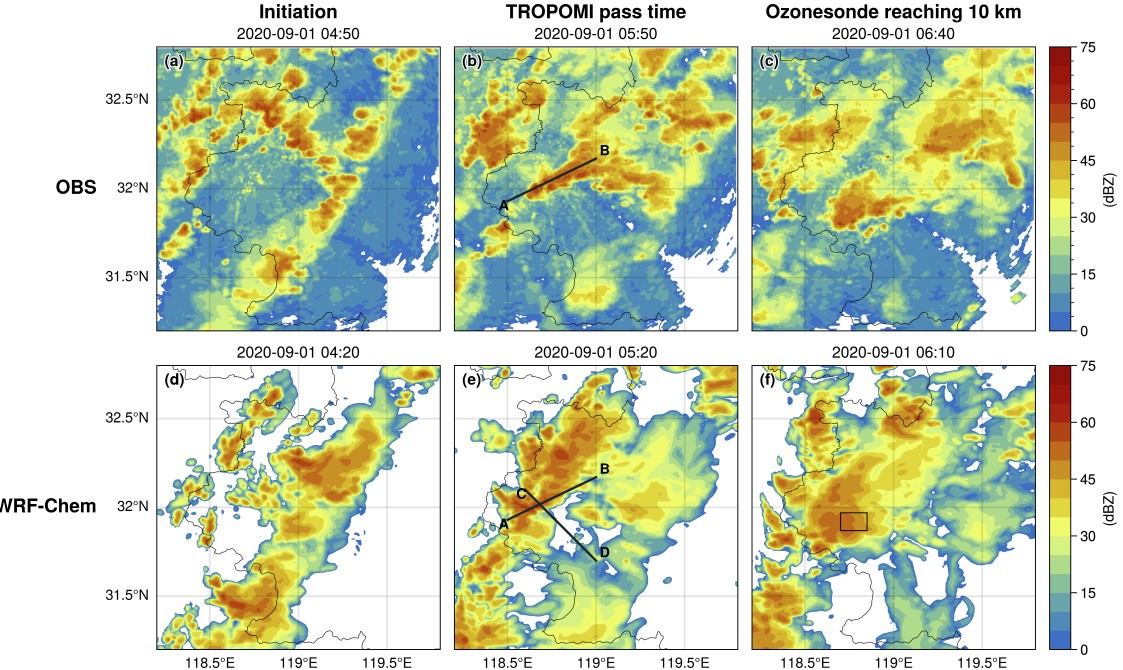

**Figure 3.** Same as Fig. 2 but for the case on 01 September 2020. The simulation time is 30 minutes ahead of each radar observation.

The UT $O_3$ increases continuously during the whole cycle of the 2020 case, but for the 2019 case, it declines at the developing stage because of the uplift of $O_3$-poor air while later it starts rising again. This phenomenon can be explained by the $O_3$ vertical cross sections at the developing stages (Fig. 4b and e). For the 2019 case, the cells of low $O_3$ concentration reach 16 km by the updraft, and then the high $O_3$ air wrapped behind the convection moves into the region. However, the observed increasing $O_3$ of the 2020 case is mostly from the vertically transported background $O_3$.

To determine the processes causing the differences between the two cases, we analyzed the outputs of mean integrated physical rates (IPR) from 10 to 14 km during the convective period (Fig. 4c and f). Generally, the opposite trend of the horizontal advection (advh) and vertical advection (advz) governs the net decrease in UT $O_3$ production rate of the 2019 case. Note that the advz contribution is negative between 10 and 11.5 km and positive between 11.5 and 13.8 km. This is due to the uplifted $O_3$-poor air and downward $O_3$-rich air caused by the stronger updraft compared with the 2020 case. As indicated by the higher $Q_v$ after convection (Fig. 1c) and tropopause height (Fig. 4b), the updraft of the 2020 case is not strong enough to wrap the stratospheric $O_3$ like the mesoscale convective system (Phoenix et al., 2020). While the dynamic processes play an important role in the $O_3$ production, the positive chemistry contribution cannot be neglected in both cases and leads to the net increase in UT $O_3$ during the convective period of 2020 case. Specifically, the chemistry increases $O_3$ in both cases and the magnitude of the effect is 5–10 times that of dynamic effects. This demonstrates the dominant chemistry role in the overall effects of convection.

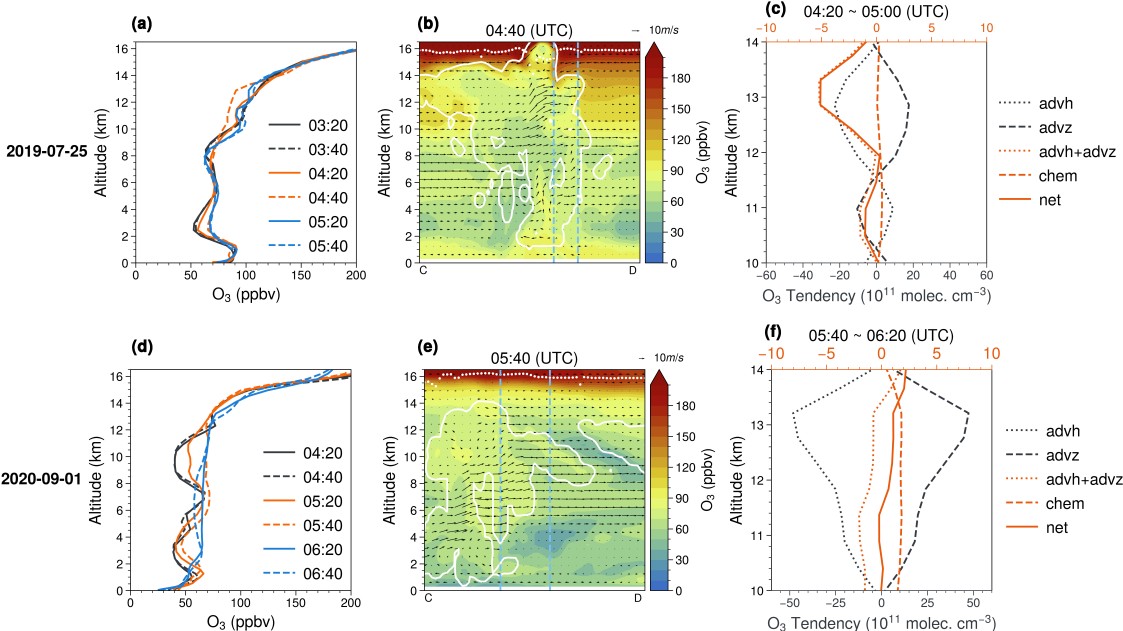

**Figure 4.** (a, d) The mean $O_3$ profiles in the regions passed by the ozonesondes at three stages: initiation (black), development (orange), and dissipation (blue). (b, e) Vertical $O_3$ distribution within the convective periods along the line crossing the convective core (Fig. 2e and Fig. 3f). The blue dashed lines stand for the boundaries of regions passed by the ozonesondes, and the lapse rate tropopause is shown as the white dots. The cloud boundaries (the sum of cloud liquid water mixing ratio [$q_{cloud}$] and ice mixing ratio [$q_{ice}$] $\geq$ 0.01 g/kg) are shown in white lines. (c, f) The vertical distributions of the $O_3$ net production rate and tendency due to horizontal advection (advh), vertical (advz) advection, and chemistry (chem) during the convective periods.

## 5   Impact of lightning $NO_x$ on the $O_3$ profile

Furthermore, the IPR outputs including LNO emission are compared with these excluding LNO to explore the effects of $LNO_x$ on $O_3$ (Table 1). The $LNO_x$ reduces the net $O_3$ production by 25 % and 40 % during the convective period and life cycle of the 2019 case, respectively. The decreased chemistry contribution is less significant ($\leq$ 1%) for the 2020 case which has a smaller lightning density near the station. Note that the $LNO_x$ can certainly enhance the downwind ozone production on the scale of days (Pickering et al., 1996; DeCaria et al., 2005). Therefore, it is necessary to estimate the $LNO_x$ PE accurately (discussed

later in Sect. 6.2).

Additionally, the convection was divided into three regions by TROPOMI data: fresh lightning , downwind of fresh lightning, and aged lightning (See Sect. 6.1 and Fig. 8 for details). Firstly, the difference of $O_3$ ($\Delta O_3$) profiles is obtained with different LNO PE assumption (Fig. 5a–c). In contrast with the net loss of ozone ($<$ 4 ppbv) over all height levels in Ott et al. (2007), the $\Delta O_3$ is mostly positive ($<$ 1 ppbv) between 2 km and 5 km and negative ($>$ -3 ppbv) between 5 km and 12 km in our

cases. The higher PE (700 mol/flash) slightly reduces the $O_3$ concentration by less than 1 ppbv at all levels compared with the

**Table 1.** Process analysis table for the mean $O_3$ integrated tendencies (10–14 km).

| Period | Time | LNO (mol/flash) | advh + advz* | chem* | net* |
|---|---|---|---|---|---|
| Life Cycle | 2019-07-25 | 0 | -3.3 (-24.6 %) | 16.7 (124.6 %) | 13.4 |
| | (03:20–05:40) | 500 | -2.3 (-28.8 %) | 10.3 (128.8 %) | 8.0 |
| | 2020-09-01 | 0 | 3.4 (9.6 %) | 32.0 (90.4 %) | 35.4 |
| | (04:20–06:40) | 500 | 4.4 (12.1 %) | 31.9 (87.8 %) | 36.3 |
| Convective Period | 2019-07-25 | 0 | -19.6 (140.0 %) | 5.6 (-40.0 %) | -14.0 |
| | (04:20–05:00) | 500 | -20.0 (114.3 %) | 2.5 (-14.3 % ) | -17.5 |
| | 2020-09-01 | 0 | -9.7 (-131.1 %) | 17.1 (231.1 % ) | 7.4 |
| | (05:40–06:20) | 500 | -10.1 (-148.5 %) | 16.9 (248.5 % ) | 6.8 |

*The unit is $10^{10}$ molec. $cm^{-3}$. The percentage is the proportion of each part in the net $O_3$ change.

default PE (500 mol/flash) and it even leads to negative $\Delta O_3$ between 2 km and 5 km downwind of fresh lightning (Fig. 5b). The maximum $O_3$ loss is between 8 km and 10 km due to the peak of $LNO_x$ (up to 2.6 ppbv) introduced in the model.

Then, the integrated reaction rate (IRR) is applied to determine the chemistry mechanism and the effect of LNO on the $O_3$ variation for two layers where the $\Delta O_3$ is opposite: 800 hPa–500 hPa ($\Delta O_3 > 0$) and 500 hPa–200 hPa ($\Delta O_3 < 0$). The
225 tropospheric $O_3$ is mainly controlled by five reaction rate terms (Pickering et al., 1990; Bozem et al., 2014):

$$\frac{d}{dt}[O_3] = k_1[NO][HO_2] + \sum_i k_i[NO][R_iO_2]$$
$$- k_3[H_2O][O(^1D)] - k_4[HO_2][O_3] - k_5[OH][O_3] \tag{4}$$

where $k_i$ is the rate coefficient of the reaction between peroxy radicals ($R_iO_2$) and NO. The time series of each contribution to $O_3$ production are illustrated in Fig. 5d–i. Overall, the time series of IRR are more variable for the 2019 case due to the
230 stronger activity as clarified in Sect. 4. The total net chemistry production of $O_3$ keeps positive for both layers. In detail, the reaction between NO and $HO_2$ always dominates the production while the oxidation of NO by $RO_2$ is about 40 %–60 % of that production. The dominant loss of $O_3$ is the photolysis (described by the reaction of $O(^1D)$ and $H_2O$) while the reaction between $O_3$ and OH is comparable during the convective period. The lowest contribution to $O_3$ loss, $O_3 + HO_2 \rightarrow OH +$ $2O_2$, is reduced during the convection because of the production of LNO, which captures the $HO_2$ reacting with $O_3$. Note that
although the increase of total IRR induced by $LNO_x$ can reach $1.36 \times 10^7$ molec. $cm^{-3}$ $s^{-1}$ and $2.60 \times 10^6$ molec. $cm^{-3}$ $s^{-1}$ in the low layer and high layer over these three regions, respectively, the net $O_3$ production actually decreases in the high layers (Fig. 5a–c) due to the combination of dynamic transport and chemical production related to $LNO_x$.

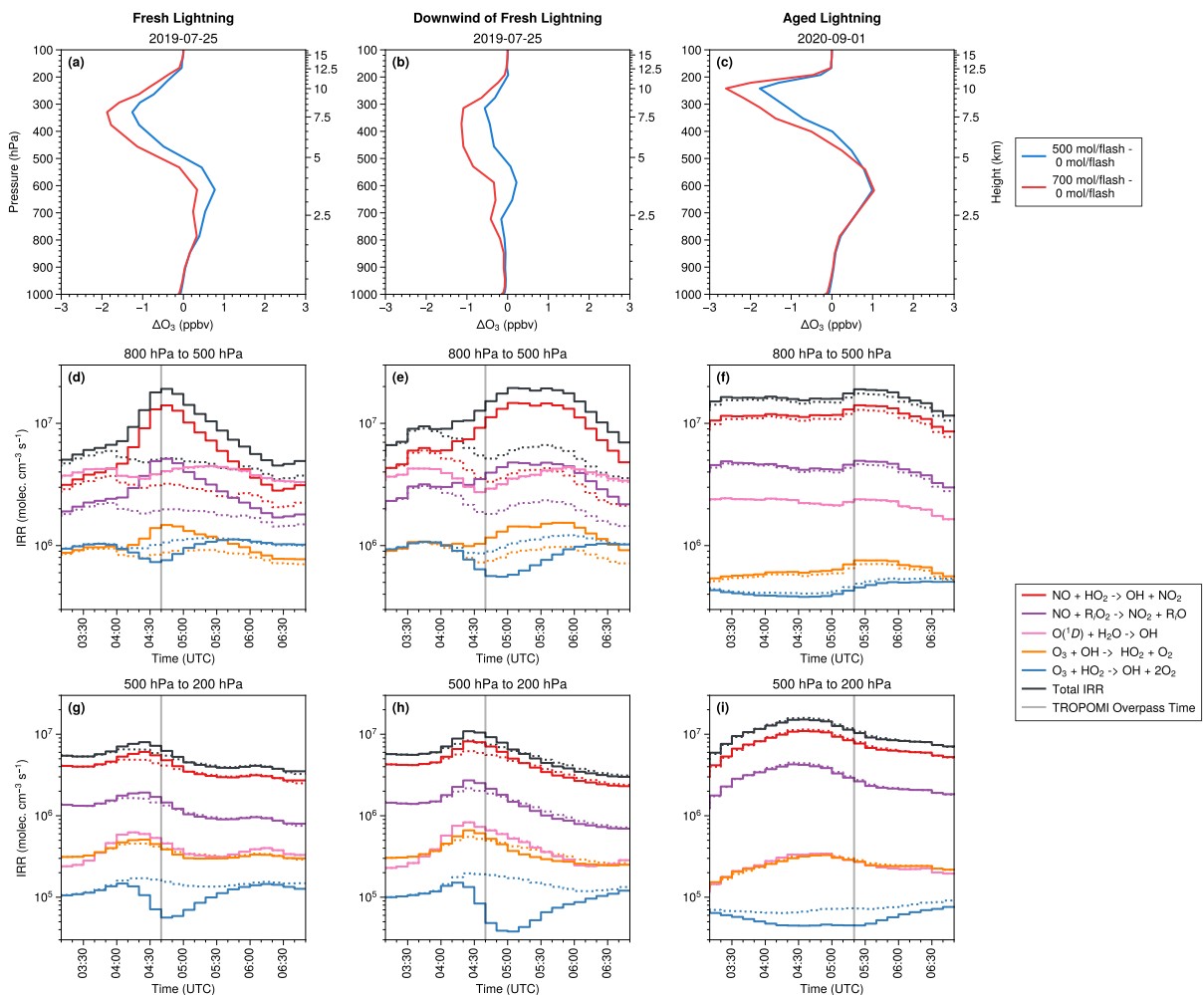

**Figure 5.** (a–c) Changes in $O_3$ profiles due to $LNO_x$ at TROPOMI overpass time in three regions (fresh lightning region, downwind of fresh lightning, and aged lightning area) as defined in Fig. 7. (d–f) Time series of the mean integrated reaction rate (IRR) between 800 hPa and 500 hPa. The legend shows detailed species and reactions. The Total IRR is the $O_3$ loss IRR subtracted from the $O_3$ production IRR (red and purple lines). The solid line shows the IRR with LNO (500 mol/flash) while the dashed line is without LNO. (g–i) Same as (d–f) but between 500 hPa and 200 hPa.

## 6 TROPOMI products over the convection

### 6.1 Relation between lightning and TROPOMI products

As the $LNO_x$ production estimation from TROPOMI depends upon the $SCD_{tropNO_2}$, we compare the $SCD_{tropNO_2}$ distributions with the observed lightning flashes (Fig. 6a–d). Although the $SCD_{tropNO_2}$ over the most active pixels is not valid due to the

detector saturation and blooming effect, the nearby or outflow regions still have useful data. While the flashes occurred less than 30 minutes before the TROPOMI overpass time in the 2019 case, the 2020 case had both fresh and aged $LNO_2$ (Fig. 6d).

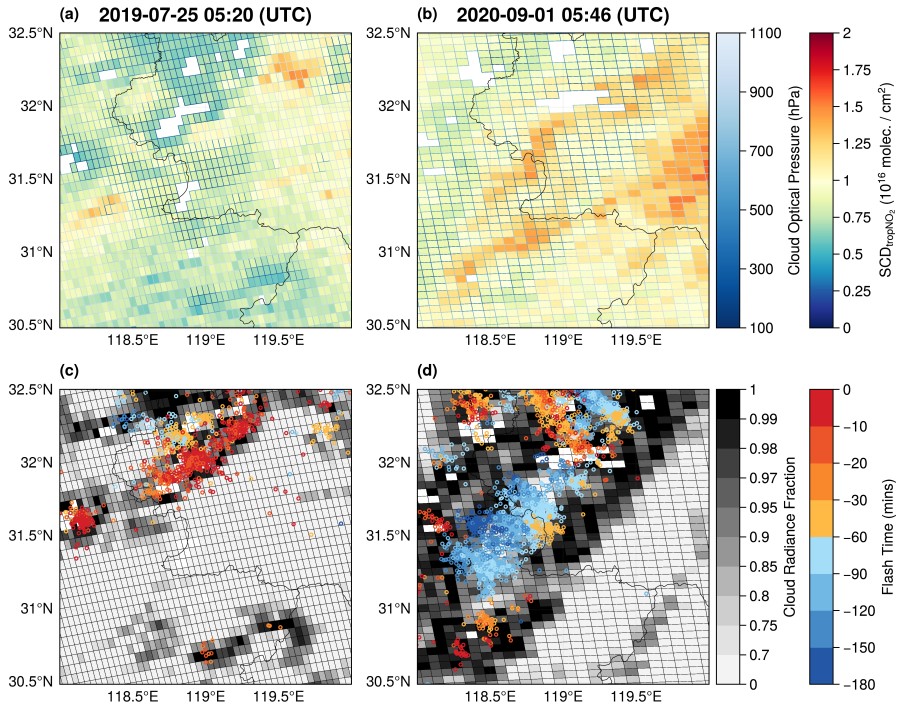

**Figure 6.** Events on 25 July 2019 (left) and 07 September 2020 (right). (a, b) The tropospheric $NO_2$ slant column density ($SCD_{tropNO_2}$, filled color) and cloud optical pressure (line color). These white grid cells stand for missing TROPOMI data (no2_scd_flag $\neq$ 0), as defined in Appendix A. The solid black border is Jiangsu province. (c, d) The cloud radiance fraction in the $NO_2$ window and flashes whose color depends on the occurring time relative to the TROPOMI overpass time.

Specifically, the $SCD_{tropNO_2}$ of the convective pixels ($f_r \geq 0.7$) is smaller than that in other regions. This is opposite to previous studies of large-scale convective systems with high flash density (Beirle et al., 2009). Four factors could lead to this unexpected result: the cloud top heights, flash counts, flash occurring time, and background $NO_2$. Either the inadequate flash or weak convection could lead to a smaller $SCD_{tropNO_2}$ over pixels with $f_r \approx 1$ because the TROPOMI can only see the $LNO_2$ above the clouds. In other words, the polluted $NO_2$ below the broken or thinner clouds is partially exposed if $f_r < 1$. The sensitivity tests of the WRF-Chem a priori $SCD_{tropNO_2}$ can help explain this phenomenon clearly (Fig. S5). The pixels of high $SCD_{tropNO_2}$ with low cloud fraction belong to background $NO_2$ pollution (Fig. S5a and e), but the $SCD_{tropNO_2}$ increased by UT $LNO_2$ is still visible compared with the lower $SCD_{tropNO_2}$ without $LNO_2$ (Fig. S5b–d and f–h).

An LNO PE upper limit of 700 mol NO/flash (Ott et al., 2010) is applied to WRF-Chem for investigating the importance of $LNO_x$ for the $AMF_{trop}$ and $AMF_{LNO_x}$ calculations. The changes in the retrieved AMFs are examined by replacing profiles in three tropospheric layers independently: middle troposphere (MT, 800 hPa to 400 hPa), upper troposphere (UT, 400 hPa to 150

hPa), and troposphere (surface to tropopause). Unless otherwise specified, the changes of AMFs are obtained by increasing $LNO_x$ in this section. Figure 7 shows that the AMF changes are mostly controlled by the $LNO_x$ in the UT layer where the detection sensitivity is high (Beirle et al., 2009; Laughner and Cohen, 2017) and the $LNO_x$ production reaches the peak (Fig. S6). While the $AMF_{LNO_x}$ decreases by 5 %–40 % for both cases, the changes of $AMF_{trop}$ ($\Delta AMF_{trop}$) are regionally specific and can be classified by the lightning activity: fresh lightning (MT $\Delta AMF_{trop} < -20$ %), downwind of fresh lightning (MT $\Delta AMF_{trop} > 20$ %), and aged lightning (UT $\Delta AMF_{trop} > 20$ %). Figure 8a illustrates the relationship between $p_{cloud}$ and $f_r$ over these three regions. The clouds are higher than 400 hPa ($p_{cloud} < 400$ hPa) and $f_r$ is larger than 0.8 over fresh lightning pixels, but both aged lightning and downwind of fresh lightning areas have clouds lower than 400 hPa. This coincides with the mean cloud pressures in Fig. S6 and explains why UT $\Delta AMF_{trop} > 20$ % exits in Fig. 7$b_i$ and $b_{iii}$, indicating the possibility of $LNO_x$ estimations over the aged lightning regions (Sect. 6.2).

As defined in Appendix B, $SCD_{tropNO_2}^{LNO_x}/SCD_{tropNO_2}^{noLNO_x}$ and $VCD_{tropNO_2}^{LNO_x}/VCD_{tropNO_2}^{noLNO_x}$ can be used to determine which parameter controls $\Delta AMF_{trop}$: enhanced a priori $SCD_{tropNO_2}$ or a priori $VCD_{tropNO_2}$ (Fig. 8b–d). Briefly, the dominant one belongs to the larger ratio. First, if the $LNO_2$ is included in the tropospheric layer of a priori $NO_2$ profiles (Fig. 7$c_i$), the $AMF_{trop}$ decreases over most fresh lightning pixels because of the increased a priori $VCD_{tropNO_2}$ (Fig. 8b). The situation is opposite for the downwind of the fresh lightning region. There the $AMF_{trop}$ is larger no matter which layer is chosen (Fig. 7$a_i$–$c_i$), because the $LNO_2$ was convected, reached above the cloud top, and led to the larger a priori $SCD_{tropNO_2}$ (Fig. 8c). Interestingly, for the 2020 case, the $AMF_{trop}$ of aged lightning pixels increases more than 50 % for the UT layer (Fig. 7$b_{iii}$). It demonstrates the important role of advected UT $LNO_2$ and that the cloud exists as a barrier, causing the difference between a priori $SCD_{tropNO_2}$ and a priori $VCD_{tropNO_2}$ (Fig. 8d). Although the difference is smaller than that of the other two regions due to the $LNO_2$ lifetime, it is still useful for retrieving the $LNO_x$. Besides, considering the region-specific $LNO_x$ effects on AMFs, we need to include the representation of $LNO_2$ in the TROPOMI $NO_2$ retrievals better, especially outflow regions. The comparisons (Fig. S6) between the TROPOMI standard $NO_2$ profiles from TM5-MP and WRF-Chem also illustrate the importance of $LNO_x$, resolved convection transport, and emissions. Aircraft observations of NO and $NO_2$ will be useful to determine the exact roles (Laughner and Cohen, 2017).

## 6.2   Estimations of $LNO_x$

Satellite observations with fresh convection are usually used to estimate the $LNO_x$ PE. However, it is difficult to apply the same method to regions with small convection like the 2019 case, because of the pixel saturation of TROPOMI and limited coverage of convective area (Fig. 6c). Instead, we focus on dissipated convection which is the southern part of the 2020 case, while the northern part has multiple missing values regardless of the adequate flashes and high cloud fractions. As shown in Fig. 9b–c, the time differences between flashes and TROPOMI overpass time are longer than 30 minutes but shorter than 3 hours. Since the lifetime of $NO_2$ is $\sim 3$ hours in or near the field of convection (Nault et al., 2016) and ranges from 2 to 12 h depending on

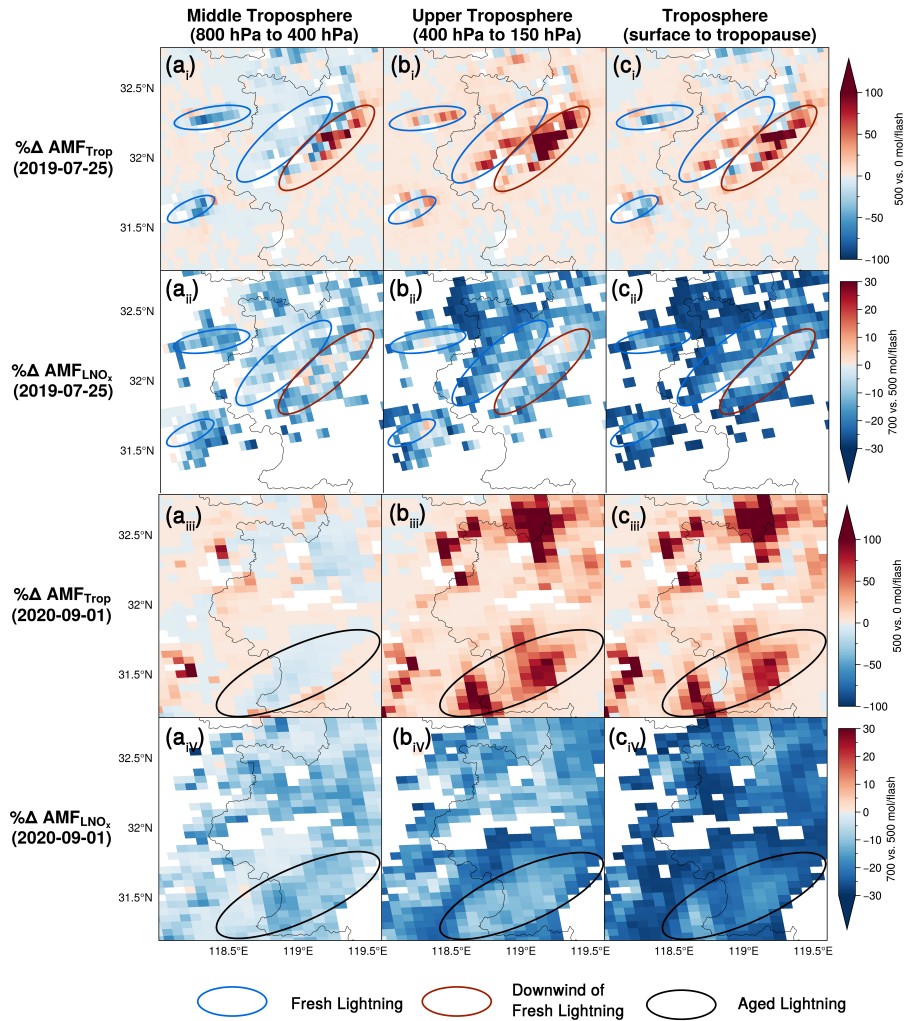

**Figure 7.** The percent differences of AMFs by replacing the a priori $NO_2$ profiles at three layers: middle troposphere (left), upper troposphere (middle), and troposphere (right). $\Delta AMF_{trop}$ is the comparison of the $AMF_{trop}$ with 500 mol NO per flash relative to 0 mol NO per flash. $\Delta AMF_{LNO_x}$ is the comparison of the $AMF_{LNO_x}$ with 700 mol NO per flash relative to 500 mol NO per flash. Three regions are annotated: fresh lightning (blue), downwind of fresh lightning (red), and aged lightning (black). Because of the quite large $AMF_{LNO_x}$ values in pixels with little lightning, $\Delta AMF_{LNO_x}$ is shown over pixels where $0 < AMF_{LNO_x} < 10$.

the convective location, these pixels can still be used for the $LNO_x$ estimation. Equation (5) is applied to determine the mean $LNO_x$ PE (mol/flash):

$$PE_{LNO_x} = \sum_p V_i A_i / \sum_N F_j e^{-(t_0 - t_j)/\tau} \tag{5}$$

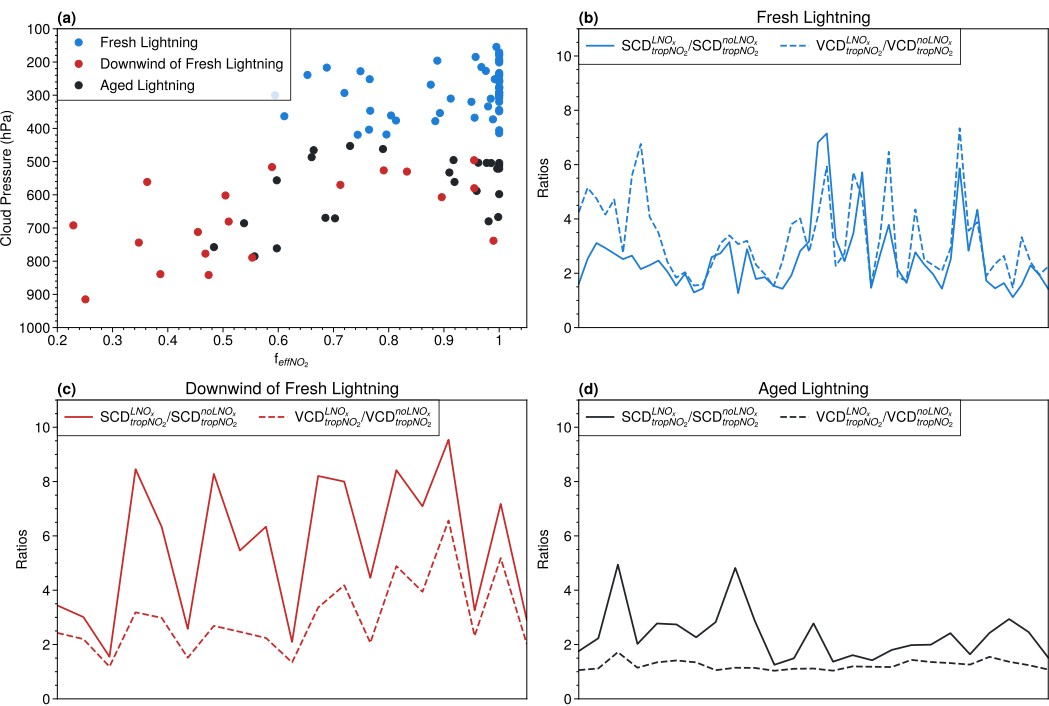

**Figure 8.** (a) The relationship between cloud pressure and cloud radiance fraction for three regions defined in Fig. 7: fresh lightning region, downwind of fresh lightning, and aged lightning area. (b–d) The a priori $\text{SCD}_{\text{tropNO}_2}^{\text{LNO}_x}/\text{SCD}_{\text{tropNO}_2}^{\text{noLNO}_x}$ and a priori $\text{VCD}_{\text{tropNO}_2}^{\text{LNO}_x}/\text{VCD}_{\text{tropNO}_2}^{\text{noLNO}_x}$ of pixels in these three regions. The $\text{LNO}_x$ superscript indicates that the a priori variable is calculated with $\text{LNO}_x$ (500 mol NO per flash) and the $\text{noLNO}_x$ superscript is without $\text{LNO}_x$.

where $p$ stands for pixels affected by $\text{LNO}_x$, $V_i$ (mol/m$^2$) is the $\text{LNO}_x$ vertical column densities ($\text{VCD}_{\text{LNO}_x} = \text{SCD}_{\text{tropNO}_2}$
/ $\text{AMF}_{\text{LNO}_x}$) over pixel $i$ with an area called $\text{A}_i$ (m$^2$), $N$ is the total number of flashes contributing to $\text{VCD}_{\text{LNO}_x}$, and the exponential component considers the lifetime of $\text{NO}_x$ for each flash ($\text{F}_j$). Specifically, $\text{t}_0$ is the time of TROPOMI overpass, $\text{t}_j$ is the time of the lightning flash, and $\tau$ represents the $\text{NO}_x$ 3-hour lifetime near convection.

As the dissipated convection produced enough lightning and the UT winds within the storm were blowing from the westnorthwest to east-southeast (Fig. 9a), the pattern of $\text{VCD}_{\text{LNO}_x}$ can still be clearly identified (dashed rectangle in Fig. 9). Fortunately, there is a low $\text{VCD}_{\text{LNO}_x}$ strip separating the northern and southern convection. With the careful selection, the $\text{LNO}_x$ PE is estimated as 60 mol $\text{NO}_x$ per flash. Although there are a few lightning flashes related to the $\text{VCD}_{\text{LNO}_x}$ is outside of the region selection, it only affects the $\text{LNO}_x$ PE by $\approx 2$ mol, which is within the uncertainty discussed below.

Following Allen et al. (2019) and Zhang et al. (2020), the uncertainty of $\text{LNO}_x$ is determined by $\text{LNO}_x$ lifetime, lightning DE, $\text{NO}/\text{NO}_2$ ratio, LNO profile, and other sources (summarized in Table 2). The lifetime ($\tau$) of $\text{NO}_2$ is replaced by 2 and 6 hours to evaluate the uncertainty as 27 % while another uncertainty is also 27 % related to lightning DE by changing the ratio of IC to CG to 2:1 and 4:1. The uncertainty caused by modeled $\text{NO}/\text{NO}_2$ ratios is assumed to be 30 % based on Allen et al.

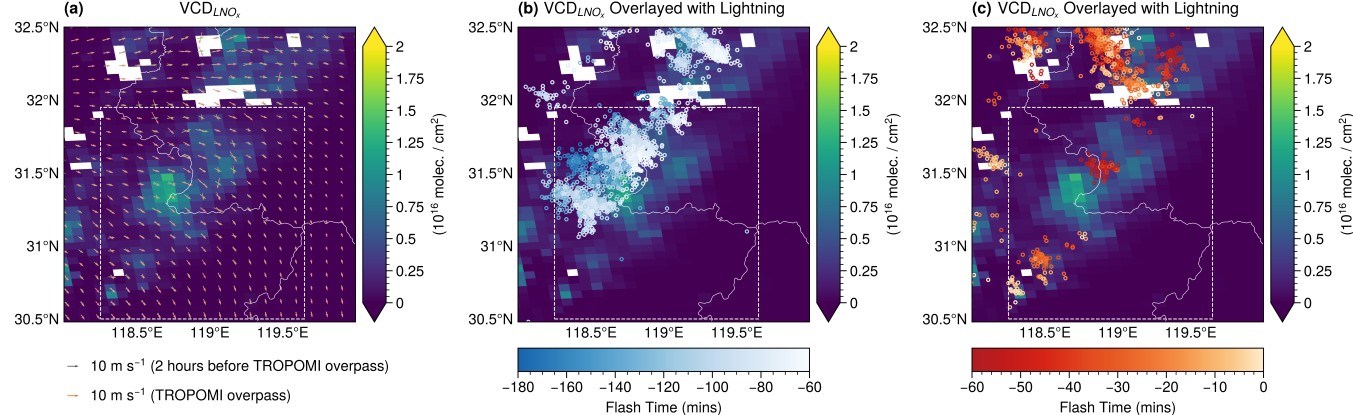

**Figure 9.** The background is the distribution of $LNO_x$ vertical column densities ($VCD_{LNO_x}$). The white rectangles are manually selected regions for the $LNO_x$ PE estimation. The overlayed wind arrows in (a) are the 500 hPa horizontal wind simulated by WRF-Chem. The lightning dots in (b) and (c) are the flashes whose color depends on the occurring time relative to the TROPOMI overpass time.

(2019) and the uncertainty related to LNO profile is 26 % by using the a priori $NO_2$ profile with 330 and 700 mol NO per flash. The uncertainty associated with the stratospheric vertical column is considered as 7 % by applying a bias of $\pm 10^{14}$ molec. cm$^2$ (van Geffen et al., 2022). The uncertainty caused by other possible error sources including systematic errors in slant columns, optical cloud pressure, and $NO_2$ redistributed by convection is difficult to quantify and assumed as 10 % following Allen et al.
(2021). Assuming no correlation between errors, the total uncertainty (56 %) is estimated as the square root of the sum of the squares of all individual uncertainties. As a result, the $LNO_x$ PE is $60 \pm 33$ mol $NO_x$ per flash. It is less than that of our previous work ($90 \pm 50$ mol $NO_x$ per flash) over the continental United States (Zhang et al., 2020) and at the lower end of $120 \pm 65$ mol $NO_x$ per flash obtained in Allen et al. (2021). Thus, more studies over China are necessary for the estimation of
region-dependent $LNO_x$ PEs.

**Table 2.** Uncertainties for the estimation of $LNO_x$ production efficiency.

| Type | Uncertainty (%) |
|---|---|
| $LNO_x$ lifetime | 27 % |
| Lightning detection efficiency | 27 % |
| NO/$NO_2$ ratio | 30 % |
| LNO profile | 26 % |
| Stratospheric vertical column | 7 % |
| Others | 10 % |
| Net | 56 % |

## 7 Conclusions

Both the 2019 and 2020 cases saw enhanced upper tropospheric (UT) $O_3$ concentrations according to the ozonesonde observations. As revealed by the WRF-Chem model, the dynamic contribution of $O_3$ variation in the 2019 case was generated by mixing with the UT $O_3$-rich air due to strong updraft, while it was caused by vertical advection of high background $O_3$ in the 2020 case. The detailed analysis of integrated physical rates shows that the dynamic processes dominate the UT $O_3$ decrease during the convective stage of both cases. However, in the convection life cycle, the contribution of chemistry reactions to the UT $O_3$ production is 5–10 times that of dynamics. Besides, the UT $O_3$ enhancement of the 2019 case decreases by 40 % if the lightning nitrogen oxides ($LNO_x$) is included in the model, indicating the importance of the $LNO_x$. Utilizing the outputs of integrated reaction rates improves our understanding of the chemical reactions and effects of lightning nitrogen oxides ($LNO_x$) on $O_3$. While the reaction of NO and $HO_2$ dominates the production of $O_3$, the $O_3$ loss processes are suppressed because the $LNO_x$ reacts with the $HO_2$ which is the reactant in the production.

The WRF-Chem results are incorporated into the retrieval algorithm to explore how $LNO_x$ affects the official TROPOMI products and prove that TROPOMI is useful for $LNO_x$ studies over small-scale convection regions. The sensitivity tests imply that air mass factors of tropospheric $NO_2$ are smaller for fresh lightning regions and the opposite for the aged lightning pixels or downwind of fresh lightning. The air mass factors of tropospheric $LNO_x$ always decrease with increased $LNO_x$. Since the $LNO_x$ affects the variation of tropospheric $NO_2$ in the outflow regions, better consideration of $LNO_x$ is essential for studies of the tropical and mid-latitude regions in summer.

Because the saturation of TROPOMI pixels and blooming effects lead to the failure of detecting $LNO_x$ over active convection, we focus on the dissipation regions where aged $LNO_x$ still exists. The production efficiency is estimated to be $60 \pm 33$ mol $LNO_x$ per flash, which is less than but still within the range of previous studies. Although the current results are limited, the technique can be implemented worldwide to region-specific $LNO_x$ retrievals. To quantify and refine the $LNO_x$ production estimates, TROPOMI data over both active and dissipated convection could provide valuable information.

*Code and data availability.* Data used are obtained from https://www.acom.ucar.edu/waccm/download.shtml (WACCM), http://meicmodel. org/ (MEIC), and https://www.acom.ucar.edu/wrf-chem/download.shtml (MEGAN). The relevant data and analysis code are hosted at https: //github.com/zxdawn/Xin_ACP_2021_Convection_Effect (Zhang, 2021a, b). The modified WRF-Chem source code is available to the public at https://github.com/zxdawn/WRF-Chem-LDA-LFR (Zhang, 2021c). The retrieval algorithm is available at https://github.com/zxdawn/ S5P-WRFChem (Zhang, 2021d).

## Appendix A: Flag definitions used in this study

The no2_scd_flag is introduced to make usage of the $NO_2$ slant column density (SCD) data easier, by gathering information from a few variables into one flag (Table A1). This flag can thus be used for filtering, though with care as it probably does not cover all possible situation. Here "$\delta$" refers to the SCD error (in mol/m$^2$) and "pqf" stands for processing quality flag.

**Table A1.** Definition of no2_scd_flag.

| Value | Meaning |
|---|---|
| -1 | no SCD value due to saturation limit exceeded, i.e. pqf=54 |
| 0 | SCD with $\delta < 3.3 \times 10^{-5}$ & no error reported |
| 1 | SCD with $\delta < 3.3 \times 10^{-5}$ & error reported: pqf=55 |
| 2 | SCD with $\delta < 3.3 \times 10^{-5}$ & other error reported, e.g. pqf=41 |
| 3 | SCD with $\delta \geq 3.3 \times 10^{-5}$ & no error reported |
| 4 | SCD with $\delta \geq 3.3 \times 10^{-5}$ & error reported: pqf=55 |
| 5 | SCD with $\delta \geq 3.3 \times 10^{-5}$ & other error reported, e.g. pqf=41 |
| FillValue | no SCD due other error (prior to the Differential Optical Absorption Spectroscopy fit) |

## Appendix B: Contributions to $\Delta\mathrm{AMF}_{\mathrm{trop}}$

In Fig. 8b–d, the contribution of $\mathrm{LNO}_x$ to $\Delta\mathrm{AMF}_{\mathrm{trop}}$ is divided into two parts: $\mathrm{SCD}_{\mathrm{tropNO_2}}^{\mathrm{LNO}_x}/\mathrm{SCD}_{\mathrm{tropNO_2}}^{\mathrm{noLNO}_x}$ and $\mathrm{VCD}_{\mathrm{tropNO_2}}^{\mathrm{LNO}_x}/\mathrm{VCD}_{\mathrm{tropNO_2}}^{\mathrm{noLNO}_x}$, where the $\mathrm{LNO}_x$ superscript indicates that the a priori variable is calculated with $\mathrm{LNO}_x$ (500 mol NO per flash) and the $\mathrm{noLNO}_x$ superscript is without $\mathrm{LNO}_x$. The two contributions are derived by taking the logarithm of Eq. (B1) and Eq. (B2) and then subtracting them into Eq. (B3). Here, several abbreviations are defined to simplify the symbols: $S$ is $\mathrm{SCD}_{\mathrm{tropNO_2}}$, $V$ is $\mathrm{VCD}_{\mathrm{tropNO_2}}$, subscript 1 is with $\mathrm{LNO}_x$, and 0 is without $\mathrm{LNO}_x$.

$$\mathrm{AMF}_1 = \frac{S_1}{V_1} \tag{B1}$$

$$\mathrm{AMF}_0 = \frac{S_0}{V_0} \tag{B2}$$

$$\begin{aligned}
\log(\mathrm{AMF}_1) - \log(\mathrm{AMF}_0) &= \log\left(\frac{S_1}{V_1}\right) - \log\left(\frac{S_0}{V_0}\right) \\
&= \log\left(\frac{S_1}{S_0}\right) - \log\left(\frac{V_1}{V_0}\right)
\end{aligned} \tag{B3}$$

Therefore, if $\frac{S_1}{S_0}$ is larger than $\frac{V_1}{V_0}$ (the solid line is higher than the dashed line in Fig. 8b–d), then $\mathrm{AMF}_1$ is larger than $\mathrm{AMF}_0$. In other words, these two variables determine how a priori $\mathrm{LNO}_x$ affects the retrieval of $\mathrm{NO}_2$.

*Author contributions.* YY directed the research and YY, XZ, and RvdA designed the research with feedback from the other co-authors; XZ, RvdA, HE, and JvG developed the retrieval algorithm; XZ modified lightning assimilation code written by YL; JLL provided guidance

and supporting data on the ENTLN data; XZ, KC, XK, ZZ, JH, CH, JZ, XY, and HC participated in the field campaigns; XZ performed simulations and analysis with the help of YY, RvdA, XK, JC, CH, and RS; XZ, YY, RvdA, and JLL interpreted the data and discussed the results. XZ drafted the manuscript with comments from the co-authors; RvdA, YY, and JLL edited the manuscript.

*Competing interests.* The authors declare that they have no conflict of interest.

*Acknowledgements.* This work is supported by the National Natural Science Foundation of China (grant nos. 91644224 and 42075067) and Postgraduate Research & Practice Innovation Program of Jiangsu Province (KYCX20_0922). We thank the Earth Networks Company for providing the Earth Networks Total Lightning Network (ENTLN) datasets. We acknowledge the use of the computational resources provided by the National Supercomputer Centre in Guangzhou (NSCC-GZ). We appreciate the discussions with Ryan M. Stauffer for ozonesonde measurements and Mary Barth for the WRF-Chem lightning $NO_x$ module. Finally, we thank all contributors of Python packages used in this paper, especially ProPlot and Satpy (Davis, 2021; Raspaud et al., 2018, 2021).

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
