# Peer review of "Influence of convection on the upper tropospheric $O_3$ and $NO_x$ budget in southeastern China"

_Atmospheric Chemistry and Physics, 2021_

## Referee Comment (RC2)

**Review:**
**Zhang et al., Influence of convection on the upper tropospheric $O_3$ and $NO_x$ budget in southeastern China**

**Summary**

The authors examine two convective cases, 25 July, 2019 and 1 September, 2020 using ozonesonde, TROPOMI $NO_2$ and lightning data. For both cases, they discuss $O_3$ distributions resulting from convection, advection, and chemistry. They also examine the effects of varying $LNO_x$ in local-scale models for quantifying $O_3$ production and for computing the air mass factors (AMFs) needed in $NO_2$ and $LNO_x$ satellite retrievals. The study is valuable in showing the variability of $O_3$ in different convective environments, and because the inclusion of more accurate local $LNO_x$ chemistry can potentially improve $NO_2$ retrievals.

This paper reads well, is well-organized and includes appropriate references. I have only a few general comments as well as technical corrections. If adequately addressed, I would recommend it for full publication.

**General comments:**

(1) Since $LNO_x$ is a key component of this study, affecting both $O_3$ chemistry and $NO_2$ and $LNO_x$ air mass factors, it would be helpful to show several examples of WRF-Chem vertical profiles of these constituents for the two cases under different PE assumptions near and downwind of convection. For example, $NO_2$, $LNO_2$ and $LNO_x$ profiles could be discussed in the context of the $O_3$ profiles of Fig. 4 and the $\Delta$AMFs of Fig. 6. If feasible, it also would be interesting to compare the profiles with the standard profiles from TM5 used in the TROPOMI data product to illustrate the importance of local effects.

(2) I find that giving the contributions of dynamic and chemical effects on O3 as percentages is confusing (abstract, text and conclusions) when the percentages are greater than 100 and the effects have opposite signs. For example, to summarize the life cycles of both cases, it would be clearer to say that the chemistry increases $O_3$ in both cases and that the magnitude of the effect is $5 - 10$ times the magnitude of dynamic effects (rather than using the $> 87\%$ figure).

(3) A substantial part of UT $NO_2$ seen in the 2020 event is likely *not* produced by the flashes counted in the region in the hour(s) immediately prior to overpass. As seen in Fig 5a, b, increased $SCD_{tropNO2}$ is visible in regions where the cloud pressures are higher and cloud

fractions are lower.  This is mentioned in lines 210 – 215 on page 12 (the relevance of Fig. S5 to this should also be made clearer). Some estimate of ambient $NO_2$ is needed so that it can be subtracted as a tropospheric background before the $LNO_x$ is computed. Studies have shown backgrounds can be substantial (e.g. Allen et al. 2019; Bucsela et al. 2019).

A related issue is the relatively small estimate of 10% for the error introduced by the stratosphere (Allen et al., 2019 - not 2021). This error assumed a tropospheric background subtraction that partially cancels stratospheric errors. Without this subtraction, the error would be larger.

(4) Fig. 6 $a_{ii} - c_{ii}$ and $a_{iv} - c_{iv}$   d show a decrease in $AMF_{LNOx}$ when $LNO_x$ is enhanced at higher altitudes, in contrast to the behavior of $AMF_{trop}$, which is consistent with Fig. 7.  Please include some words qualitatively discussing the behavior of $AMF_{LNOx}$.

(5) In Figure 5f, why wasn't the northern part of the region included in the $LNO_x$ analysis (section 5.2)?  There appear to be adequate flashes there, along with $LNO_x$ and a high cloud fraction.  The southern/southeastern regions include areas of low cloud fraction that could potentially contaminate the measurements with anthropogenic $NO_2$. Also, if winds are from the WNW, shouldn't the flash-counting window be displaced WNW of the $LNO_x$ window?

**Specific comments and technical corrections:**

(1) Page 2, line 49:  "We apply new a priori $NO_2$..."

(2) Page 3, line 57: "...near the airmass convection that developed on 25..."

(3) Page 3, line 64: "...the observed difference of more than 65%."

(4) Page 5, line 84: "...with a constant IC/CG ratio of 3:1 based on Wu et al..."

(5) Page 5, lines 81 – 85:  Please add some detail on how the 3 datasets were merged. Was CNDLN used to estimate a DE for ENTLN and WWLLN?

(6) Page 6, line 105: Please add a similar equation for $AMF_{trop}$, since it is used in section 5.1

(7) Page 7, line 161: "The squall line on 1 September, 2020 was born..."

(8) Page 10, Fig. 4 caption: "The vertical distributions of the $O_3$ net production rate and tendency..."

(9) Page 10, line 198: Regarding "...less significant ($< 1\%$)..."  From Table 1, I estimate that for the 2020 case, changes in chemistry affect net $O_3$ production by 0.3% and ~3% during the life cycle and convective period, resp.

(10) P10, line 201: "...can certainly enhance the downwind ozone production on the scale of days..."

(11) Page 12, Fig. 5 caption: Please state white grid cells are for missing TROPOMI data (no2_scd_flag > 0 ?).

(12) Page 13, lines 218-219: "…middle troposphere (MT, 800 hPa to 400 hPa), upper troposphere (UT 400 hPa to 150 hPa)…"

(13) Page 13, line 220-221: "...Figure 6 shows that the AMF changes...". Also, Beirle et al. (2009) noted a *decrease* in sensitivity in the UT due to the $NO_2/NO_x$ branching ratio.

(14) Page 13, line 226: "...UT $\Delta AMF_{trop} > 20$ % exists in Fig. $6b_i$ and $b_{iii}$,"

(15) Page 14, Fig 6 caption:

"...is the $AMF_{trop}$ with 500 mol NO per flash relative to 0 mol NO per flash"

"...is the $AMF_{LNOx}$ with 700 mol NO per flash relative to 500 mol NO per flash"

Also, it would help to see the ovals overplotted on all figures for easier comparison.

(16) Page 15, Fig 7: Please label the x-axes in (b), (c) and (d)

Supplement:

(17) Figure S1, caption: "…WRF-Chem simulations for the 2019 and 2020 cases."

(18) Figure S2: The times/dates in the legend of (a) are not correct

(19) Figure S3, caption: "…in Fig. 2 for 25 July, 2019."

---

## Author Comment (AC1)

**Influence of convection on the upper tropospheric $O_3$ and $NO_x$ budget in southeastern China**

**Response to Anonymous Referee #1**

Xin Zhang, Yan Yin, Ronald van der A, Henk Eskes, Jos van Geffen, Yunyao Li, Xiang Kuang, Jeff L. Lapierre, Kui Chen, Zhongxiu Zhen, Jianlin Hu, Chuan He, Jinghua Chen, Rulin Shi, Jun Zhang, Xingrong Ye, and Hao Chen

February 4, 2022

We thank the reviewer for his/her positive comments and very careful reading of our article. The individual corrections suggested are addressed below. The reviewer's comments will be shown in red, our response in blue, and changes made to the paper are shown in **black** block quotes. Unless otherwise indicated, page and line numbers correspond to the original manuscript. Figures, tables, or equations referenced as "R$n$" are numbered within this response; if these are used in the changes to the paper, they will be replaced with the proper number in the revised version.

1) Line 18: Huntrieser et al. (2016) measured ozone gradients in thunderstorms. Please cite and explain their results here. Mention again in lines 36-40.

Huntrieser, H., Lichtenstern, M., Scheibe, M., Aufmhoff, H., Schlager, H., Pucik, T., ... & Barth, M. C. (2016). On the origin of pronounced O3 gradients in the thunderstorm outflow region during DC3. Journal of Geophysical Research: Atmospheres, 121(11), 6600-6637.

Thanks for the reference and we have added it. Because the conclusion of Huntrieser et al. (2016) is similar to Phoenix et al. (2020), we combined them in the same sentence.

> Meanwhile, the vertical profiles of trace gases are reshaped by the updraft and downdraft on timescales of hours (**Huntrieser et al., 2016; Barth et al., 2019**).
>
> ... As revealed in the **Deep Convective Clouds and Chemistry 2012 Studies (DC3)** and mesoscale convective system simulations, the compensation of subsidence and differential advection beneath the convective core can lead to the anvil wrapping effects (**Huntrieser et al., 2016; Phoenix et al., 2020**).

Lines 19 - 24: Murray et al. (2012) and Gordillo-Vázquez et al. (2019) used global models to study the relationship between LNOx and other trace gases. Please cite here.

Murray, L. T., Jacob, D. J., Logan, J. A., Hudman, R. C., & Koshak, W. J. (2012). Optimized regional and interannual variability of lightning in a global chemical transport model constrained by LIS/OTD satellite data. Journal of Geophysical Research: Atmospheres, 117(D20).

Gordilloâ Vázquez, F. J., Pérezâ Invernón, F. J., Huntrieser, H., & Smith, A. K. (2019). Comparison of six lightning parameterizations in CAM5 and the impact on global atmospheric chemistry. Earth and Space Science, 6(12), 2317-2346.

Added.

Several lightning parameterizations have been developed for chemistry transport and climate models to evaluate the relationship between $LNO_x$ and other trace gases (Murray et al., 2012; Gordillo-Vázquez et al., 2019; Luhar et al., 2021; Pérez-Invernón et al., 2021).

Lines 43-44: Please give a brief explanation on why thunderstorms and lightning have increased due to urbanization. Is it because of the effect of aerosols on cloud electrification [Tao et al. (2012), Pérez-Invernón et al. (2021)]?

Tao, W. K., Chen, J. P., Li, Z., Wang, C., & Zhang, C. (2012). Impact of aerosols on convective clouds and precipitation. Reviews of Geophysics, 50(2).

Pérez-Invernón, F. J., Huntrieser, H., Gordillo-Vázquez, F. J., & Soler, S. (2021). Influence of the COVID-19 lockdown on lightning activity in the Po Valley. Atmospheric Research, 263, 105808.

Thanks for the suggestion. Yang and Li (2014) (cited in the manuscript) suggests a large impact of aerosols on thunderstorm activities. We have added it.

Little is known about the role of convection in southeastern China (Murray, 2016; Guo et al., 2017), where thunderstorm and lightning have increased significantly by urbanization during recent decades **(Yang and Li, 2014; Pérez-Invernón et al., 2021). This is likely due to the increasing aerosol concentration, which can invigorate storms in a moist and convectively unstable environment (Koren et al., 2008; Rosenfeld et al., 2008; Tao et al., 2012)**.

Line 46: Although described in other part of the manuscript, it could be useful for the reader giving a reference for TROPOMI here, as it is the first time it appears.

Added.

For the first time, the TROPOspheric Monitoring Instrument **(TROPOMI; Veefkind et al., 2012)** $NO_2$ observations are used to identify $LNO_x$ PEs in southeastern China.

Line 74: intro-cloud -> intra-cloud

Fixed.

... while ENTLN and WWLLN detect both **intra-cloud** (IC) and CG flashes with specific detection frequency ...

Section 2.2: The main weakness of the manuscript is the poor estimation on the DE of the employed lightning systems. A reliable estimation on the total number of lightning flashes is needed to calculate the LNOx. Would it be possible combining the lightning data from the used lightning dataset with lightning data from ISS-LIS in order to evaluate the combinience of using the 3:1 ratio? ISS-LIS reports IC+CG lightning. The DE of ISS-LIS is well established and quite constant all around the world [Blakeslee et al. (2020)].

Blakeslee, R. J., Lang, T. J., Koshak, W. J., Buechler, D., Gatlin, P., Mach, D. M., ... & Christian, H. (2020). Three years of the lightning imaging sensor onboard the international space station: Expanded global coverage and enhanced applications. Journal of Geophysical Research: Atmospheres,125(16), e2020JD032918.

Yes, that would be a better method for total lightning networks. Because the CLDN only detects CG flashes and the IC DE of ENGLN is low in China, we have decided to use the constant IC:CG ratio from Wu et al. (2016) and Bandholnopparat et al. (2020). If there are more total lightning networks deployed in China, we will use that for future studies with a better DE estimation. The uncertainty of $LNO_x$ PE estimation caused by DE is evaluated by changing the ratio to 2:1 and 4:1.

Although CG detection efficiency of ENGLN is not known for this region due to a lack of validation data, merging these three datasets should provide a sufficiently high CG flash detection efficiency for this

analysis. **Because the IC DE of all these lightning data is low in China, we conservatively use the merged CG data with a constant IC/CG ratio of 3:1 based on Wu et al. (2016) and Bandholnopparat et al. (2020). IC data will become more accurate if more Chinese total lightning networks, such as Beijing Lightning Network (BLNET; Srivastava et al., 2017), are available**.

Equation (1): I think a more detailed explanation on eq. (1) is needed.

Added.

We replaced the tropospheric AMF ($AMF_{trop}$) with a new AMF called AMF for $LNO_x$ ($AMF_{LNO_x}$) to derive the tropospheric $LNO_x$ vertical column density ($VCD_{LNO_x}$). The concept of $AMF_{LNO_x}$ inherits from the $AMF_{trop}$ derived by a function of several parameters (solar zenith angle, viewing zenith angle, relative azimuth angle, surface albedo, surface pressure, cloud fraction, cloud height, and a priori trace gas profile). **Briefly, the numerator is the modeled tropospheric $NO_2$ slant column density ($SCD_{tropNO_2}$) and the denominator is the modeled VCD ($VCD_{NO_2}$ or $VCD_{LNO_x}$). In detail, these two AMFs can be calculated as**:

$$AMF_{\text{Trop}} = \frac{(1 - f_{effNO_2}) \int_{p_{surf}}^{p_{tp}} w_{clear}(p) NO_2(p)\, dp + f_{effNO_2} \int_{p_{cloud}}^{p_{tp}} w_{cloudy}(p) NO_2(p)\, dp}{\int_{p_{surf}}^{p_{tp}} NO_2(p)\, dp} \quad \text{(R1)}$$

$$AMF_{\text{LNO}_x} = \frac{(1 - f_{effNO_2}) \int_{p_{surf}}^{p_{tp}} w_{clear}(p) NO_2(p)\, dp + f_{effNO_2} \int_{p_{cloud}}^{p_{tp}} w_{cloudy}(p) NO_2(p)\, dp}{\int_{p_{surf}}^{p_{tp}} LNO_x(p)\, dp} \quad \text{(R2)}$$

where $p_{surf}$ is the surface pressure, $p_{tp}$ is the tropopause pressure, $p_{cloud}$ is the cloud optical pressure, $f_{effNO_2}$ is the effective cloud fraction in the $NO_2$ window, $w_{clear}$ and $w_{cloudy}$ are respectively the pressure-dependent scattering weights from the lookup table (Lorente et al., 2017) for clear and cloudy parts, and $NO_2(p)$ is the $NO_2$ vertical profile simulated by WRF-Chem. Besides, $LNO_x(p)$ is the $LNO_x$ vertical profile calculated by the difference of vertical profile between WRF-Chem simulations with and without lightning.

Lines 174–176: Ripoll et al. (2014) developed a detailed chemistry model for lightning channel including ozone. It would be beneficial for the manuscript citing it at this point.
Ripoll, J. F., Zinn, J., Jeffery, C. A., & Colestock, P. L. (2014). On the dynamics of hot air plasmas related to lightning discharges: 1. Gas dynamics. Journal of Geophysical Research: Atmospheres, 119(15), 9196-9217.

Thanks for your suggestion. It is a valuable model. Added.

Only the first two factors are discussed in detail in Sect. 4, as lightning $O_3$ is beyond the scope of this study and still uncertain as shown by limited observations and model simulations **(Morris et al., 2010; Ripoll et al., 2014)**.

Line 246: There is still a significant uncertainty in the lifetime of NO2 in or near the field of convection [e.g., Beirle et al. (2010), Pickering et al. 2016 ...]. According to literature, the lifetime can vary between 3 h and 2 days. Although it is mentioned below, please mention here.

Added.

Since the lifetime of $NO_2$ is $\sim 3$ hours in or near the field of convection (Nault et al., 2016) **and ranges from 2 to 12 h depending on the convective location**, these pixels can still be used for the $LNO_x$ estimation.

Lines 257-263: Comparing the sources of uncertainty in the text is hard. A Table for the uncertainties could be useful.

Yes, it can make it easier to read. Added (Table R1).

**Table R1 .** Uncertainties for the estimation of $LNO_x$ production efficiency.

[revised manuscript text omitted]

---

## Author Comment (AC2)

**Influence of convection on the upper tropospheric $O_3$ and $NO_x$ budget in southeastern China**

**Response to Anonymous Referee #3**

Xin Zhang, Yan Yin, Ronald van der A, Henk Eskes, Jos van Geffen, Yunyao Li, Xiang Kuang, Jeff L. Lapierre, Kui Chen, Zhongxiu Zhen, Jianlin Hu, Chuan He, Jinghua Chen, Rulin Shi, Jun Zhang, Xingrong Ye, and Hao Chen

February 4, 2022

We thank the reviewer for his/her positive comments and very careful reading of our article. The individual corrections suggested are addressed below. The reviewer's comments will be shown in **red**, our response in **blue**, and changes made to the paper are shown in **black** block quotes. Unless otherwise indicated, page and line numbers correspond to the original manuscript. Figures, tables, or equations referenced as "R$n$" are numbered within this response; if these are used in the changes to the paper, they will be replaced with the proper number in the revised version.

**General Comments**

(1) Since $LNO_x$ is a key component of this study, affecting both $O_3$ chemistry and $NO_2$ and $LNO_x$ air mass factors, it would be helpful to show several examples of WRF-Chem vertical profiles of these constituents for the two cases under different PE assumptions near and downwind of convection. For example, $NO_2$, $LNO_2$ and $LNO_x$ profiles could be discussed in the context of the $O_3$ profiles of Fig. 4 and the $\Delta$AMFs of Fig. 6. If feasible, it also would be interesting to compare the profiles with the standard profiles from TM5 used in the TROPOMI data product to illustrate the importance of local effects.

Thanks for your suggestions. We have added another short section to discuss the effects of $LNO_x$ on $O_3$ profile using integrated reaction rate (IRR) results. Beside, the TM5, $NO_2$, $LNO_2$ and $LNO_x$ profiles are also added to the supplement.

**Impact of lightning $NO_x$ on the $O_3$ profile**

[revised manuscript text omitted]

(2) I find that giving the contributions of dynamic and chemical effects on $O_3$ as percentages is confusing (abstract, text and conclusions) when the percentages are greater than 100 and the effects have opposite signs. For example, to summarize the life cycles of both cases, it would be clearer to say that the chemistry increases $O_3$ in both cases and that the magnitude of the effect is 5–10 times the magnitude of dynamic effects (rather than using the > 87% figure).

Yes, the percentage changes are confusing. We have modified them according to your advice.

**Abstract**

... During the whole convection life cycle, the UT $O_3$ production is driven by the chemistry (5–10 times the magnitude of dynamic contribution) and reduced by the $LNO_x$ ($-40$ %).

**Convection impacts**

... While the dynamic processes play an important role in the $O_3$ production, the positive chemistry contribution cannot be neglected in both cases and leads to the net increase in UT $O_3$ during the convective period of 2020 case. Specifically, the chemistry increases $O_3$ in both cases and the magnitude of the effect is 5–10 times that of dynamic effects. This demonstrates the dominant chemistry role in the overall effects of convection.

**Conclusions**

[Figure]

**Figure R1 .** (a–c) Changes in O$_3$ profiles due to LNO$_x$ at TROPOMI overpass time in three regions (fresh lightning region, downwind of fresh lightning, and aged lightning area) as defined in Fig. 7. (d–f) Time series of the mean integrated reaction rate (IRR) between 800 hPa and 500 hPa. The legend shows detailed species and reactions. The Total IRR is the O$_3$ loss IRR subtracted from the O$_3$ production IRR (red and orange lines). The solid line shows the IRR with LNO (500 mol/flash) while the dashed line is without LNO. (g–i) Same as (d–f) but between 500 hPa and 200 hPa.

... The detailed analysis of integrated physical rates shows that the dynamic processes dominate the UT O$_3$ decrease during the convective stage of both cases. However, in the convection life cycle, the contribution of chemistry reactions to the UT O$_3$ production is 5–10 times that of dynamics.

(3) A substantial part of UT NO$_2$ seen in the 2020 event is likely not produced by the flashes counted in the region in the hour(s) immediately prior to overpass. As seen in Fig. 5a, b, increased SCD$_{tropNO_2}$ is visible in regions where the cloud pressures are higher and cloud fractions are lower. This is mentioned in lines 210 – 215 on page 12 (the relevance of Fig. S5 to this should also be made clearer). Some estimate of ambient NO$_2$ is needed so that it can be subtracted as a tropospheric background before the LNO$_x$ is computed. Studies have shown backgrounds can be substantial (e.g. Allen et al. 2019; Bucsela et al. 2019). A related issue is the relatively small estimate of 10% for the error introduced by the stratosphere (Allen et al.,

[Figure]

**Figure R2 .** Profiles with different lightning NO productions at TROPOMI overpass time over three regions (fresh lightning, downwind of fresh lightning, and aged lightning). (a–c) The $NO_2$ profiles compared with the official TM5 a priori $NO_2$ profile. (d–f) The lightning $NO_2$ and $NO_x$ profiles. The blue dashed line is the cloud optical pressure detected by TROPOMI.

2019 - not 2021). This error assumed a tropospheric background subtraction that partially cancels stratospheric errors. Without this subtraction, the error would be larger.

    1. Thanks, we have clarified the relevance of these figures for the ambient $NO_2$.
    2. Note that the ambient $NO_2$ has been considered in the numerator of $AMF_{LNO_x}$:

$$AMF_{\text{LNO}_x} = \frac{(1 - f_{effNO_2}) \int_{p_{surf}}^{p_{tp}} w_{clear}(p) NO_2(p) \, dp + f_{effNO_2} \int_{p_{cloud}}^{p_{tp}} w_{cloudy}(p) NO_2(p) \, dp}{\int_{p_{surf}}^{p_{tp}} LNO_x(p) \, dp} \qquad \text{(R2)}$$

    This is different from that of Pickering et al. (2016), Allen et al. (2019), and Bucsela et al. (2019). We have added the comparison in Sect. 2.3 (TROPOMI Data).
    3. For the stratospheric errors, we have estimated the uncertainty as 9 % by applying a bias of $\pm 10^{14}$ molec. $cm^2$ (van Geffen et al., 2021). As mentioned in Bucsela et al. (2019), if they do not consider the background subtraction, "the stratospheric bias would have a >90 % effect on PE". Therefore, the low uncertainty (9 %) related to the stratospheric vertical column indicates that our $AMF_{LNO_x}$ works well.

**Relation between lightning and TROPOMI products**

    ... Either the inadequate flash or weak convection could lead to a smaller $SCD_{tropNO_2}$ over pixels with $f_{effNO_2} \approx 1$ because the TROPOMI can only see the $LNO_2$ above the clouds. In other words, the polluted $NO_2$ below the broken or thinner clouds is partially exposed if $f_{effNO_2} < 1$. The sensitivity tests of

the WRF-Chem a priori $SCD_{tropNO_2}$ can help explain this phenomenon clearly (Fig. S5). The pixels of high $SCD_{tropNO_2}$ with low cloud fraction belong to background $NO_2$ pollution (Fig. S5a and e), but the $SCD_{tropNO_2}$ increased by UT $LNO_2$ is still visible compared with the lower $SCD_{tropNO_2}$ without $LNO_2$ (Fig. S5b–d and f–h).

**TROPOMI data**

... In comparison with this study, Pickering et al. (2016), Allen et al. (2019), Bucsela et al. (2019), and Allen et al. (2021) defined another $AMF_{LNO_x}$ to convert $SCD_{tropNO_2}$ to the tropospheric $NO_x$ vertical column density ($VCD_{NO_x}$). Then, their $VCD_{LNO_x}$ can be calculated as 1) the slope of the regression between $VCD_{NO_x}$ and flashes or 2) the $VCD_{NO_x}$ subtracted by a tropospheric $NO_x$ background. Because our $AMF_{LNO_x}$ converts the $SCD_{tropNO_2}$ to $VCD_{LNO_x}$ directly, the additional estimation of background $NO_2$ is not needed for calculating $LNO_x$ PE in Sect. 6.2.

**Estimations of $LNO_x$**

... Following Allen et al. (2019) and Zhang et al. (2020), the uncertainty of $LNO_x$ is determined by $LNO_x$ lifetime, lightning DE, $NO/NO_2$ ratio, LNO profile, and other sources (summarized in Table R2). The lifetime ($\tau$) of $NO_2$ is replaced by 2 and 6 hours to evaluate the uncertainty as 27 % while another uncertainty is also 27 % related to lightning DE by changing the ratio of IC to CG to 2:1 and 4:1. The uncertainty caused by modeled $NO/NO_2$ ratios is assumed to be 30 % based on Allen et al. (2019) and the uncertainty related to LNO profile is 26 % by using the a priori $NO_2$ profile with 330 and 700 mol NO per flash. The uncertainty associated with the stratospheric vertical column is considered as 7 % by applying a bias of $\pm 10^{14}$ molec. $cm^2$ (van Geffen et al., 2021). The uncertainty caused by other possible error sources including systematic errors in slant columns, optical cloud pressure, and $NO_2$ redistributed by convection is difficult to quantify and assumed as 10 % following Allen et al. (2021). Assuming no correlation between errors, the total uncertainty (56 %) is estimated as the square root of the sum of the squares of all individual uncertainties. As a result, the $LNO_x$ PE is $60 \pm 33$ mol $NO_x$ per flash. It is less than that of our previous work ($90 \pm 50$ mol $NO_x$ per flash) over the continental United States (Zhang et al., 2020) and at the lower end of $120 \pm 65$ mol $NO_x$ per flash obtained in Allen et al. (2021). Thus, more studies over China are necessary for the estimation of region-dependent $LNO_x$ PEs.

**Table R2 .** Uncertainties for the estimation of $LNO_x$ production efficiency.

| Type | Uncertainty (%) |
|---|---|
| $LNO_x$ lifetime | 27 % |
| Lightning detection efficiency | 27 % |
| $NO/NO_2$ ratio | 30 % |
| LNO profile | 26 % |
| Stratospheric vertical column | 7 % |
| Others | 10 % |
| Net | 56 % |

(4) Fig. 6 $a_{ii}$ — $c_{ii}$ and $a_{iv} - c_{iv}$ show a decrease in $AMF_{LNO_x}$ when $LNO_x$ is enhanced at higher altitudes, in contrast to the behavior of $AMF_{trop}$, which is consistent with Fig. 7. Please include some words qualitatively discussing the behavior of $AMF_{LNO_x}$.

Added.

While the $AMF_{LNO_x}$ **decreases by 5 %–40 %** for both cases, the changes of $AMF_{trop}$ ($\Delta AMF_{trop}$) are regionally specific and can be classified by the lightning activity: fresh lightning (MT $\Delta AMF_{trop} < -20$ %), downwind of fresh lightning (MT $\Delta AMF_{trop} > 20$ %), and aged lightning (UT $\Delta AMF_{trop} > 20$ %).

(5) In Figure 5f, why wasn't the northern part of the region included in the $LNO_x$ analysis (section 5.2)? There appear to be adequate flashes there, along with $LNO_x$ and a high cloud fraction. The southern/southeastern regions include areas of low cloud fraction that could potentially contaminate the measurements with anthropogenic $NO_2$. Also, if winds are from the WNW, shouldn't the flash-counting window be displaced WNW of the $LNO_x$ window?

We had decided to include both parts in the analysis at the first time. However, there are many missing data in the northern convection. As replied in (3), the anthropogenic $NO_2$ has been included in the retrieval algorithm, which has also been applied in Zhang et al. (2020). We have fixed a bug of deriving the $VCD_{LNO_x}$ and modified the selected region with clearer statements. The figure has been re-plotted and the $LNO_x$ PE is estimated as $60 \pm 33$ mol $NO_x$ per flash.

As the dissipated convection produced enough lightning and the UT winds within the storm were blowing from the west-northwest to east-southeast (Fig. R3a), the pattern of $VCD_{LNO_x}$ can still be clearly identified (dashed rectangle in Fig. R3). Fortunately, there is a low $VCD_{LNO_x}$ strip separating the northern and southern convection. With the careful selection, the $LNO_x$ PE is estimated as 60 mol $NO_x$ per flash. Although there are a few lightning flashes related to the $VCD_{LNO_x}$ is outside of the region selection, it only affects the $LNO_x$ PE by $\approx 2$ mol, which is within the uncertainty discussed below.

[Figure]

**Figure R3 .** The background is the distribution of $LNO_x$ vertical column densities ($VCD_{LNO_x}$). The white rectangles are manually selected regions for the $LNO_x$ PE estimation. The overlayed wind arrows in (a) are the 500 hPa horizontal wind simulated by WRF-Chem. The lightning dots in (b) and (c) are the flashes whose color depends on the occurring time relative to the TROPOMI overpass time.

**Specific comments and technical corrections**

1) Page 2, line 49: "We apply new a priori NO2..."

Thanks, fixed.

We apply **new a priori NO$_2$** profiles into the retrieval algorithm to explore the sensitivity of AMFs to LNO$_x$.

2) Page 3, line 57: "...near the airmass convection that developed on 25..."

Fixed.

Three Institute of Atmospheric Physics (IAP) ozonesondes had been launched near the airmass convection **that** developed on 25 July 2019.

3) Page 3, line 64: "...the observed difference of more than 65%."

Fixed.

therefore the daily variation cannot explain the observed difference of **more than 65 %**.

4) Page 5, line 84: "...with a constant IC/CG ratio of 3:1 based on Wu et al..."

Fixed.

Because the IC DE of all these lightning data is low in China, we conservatively used the merged CG data **with a constant IC/CG ratio of 3:1 based on** ...

5) Page 5, lines 81 – 85: Please add some detail on how the 3 datasets were merged. Was CNDLN used to estimate a DE for ENTLN and WWLLN?

Added.

The detection efficiency (DE) of cloud-to-ground (CG) flashes is about 90 % for the CNLDN data in Jiangsu province (Li et al., 2017) while ENTLN and WWLLN detect both intro-cloud (IC) and CG flashes with specific detection frequency (1 Hz–12 MHz for ENTLN and 3–30 kHz for WWLLN). In the ENTLN data, groups of pulses are classified as a flash if they are within 700 ms and 10 km. Both strokes and lightning flashes composed of one or more strokes are included in the preprocessed data obtained from the ENTLN. The detailed processing algorithm of the WWLLN is given by Rodger et al. (2004). The WWLLN strokes and pulses are combined with ENTLN into one dataset (ENGLN) within 10 km and 0.7 s as mentioned in Virts and Goodman (2020). **To increase the lightning data coverage in our study, the CG flashes of ENGLN and CNLDN datasets are combined using spatial and temporal clustering criteria of 10 km and 0.5 s (Zhao et al., 2020). Although CG detection efficiency of ENGLN is not known for this region due to a lack of validation data, merging these three datasets should provide a sufficiently high CG flash detection efficiency for this analysis. Because the IC DE of all these lightning data is low in China, we conservatively use the merged CG data with a constant IC/CG ratio of 3:1 based on Wu et al. (2016) and Bandholnopparat et al. (2020). IC data will become more accurate if more Chinese total lightning networks, such as Beijing Lightning Network (BLNET; Srivastava et al., 2017), are available**.

6) Page 6, line 105: Please add a similar equation for AMF$_{trop}$, since it is used in section 5.1.

Added.

We replaced the tropospheric AMF (AMF$_{trop}$) with a new AMF called AMF for LNO$_x$ (AMF$_{LNO_x}$) to derive the tropospheric LNO$_x$ vertical column density (VCD$_{LNO_x}$). The concept of AMF$_{LNO_x}$ inherits from the AMF$_{trop}$ derived by a function of several parameters (solar zenith angle, viewing zenith angle, relative azimuth angle, surface albedo, surface pressure, cloud fraction, cloud height, and a priori trace gas profile). Briefly, the numerator is the modeled tropospheric NO$_2$ slant column density (SCD$_{tropNO_2}$) and the denominator is the modeled VCD (VCD$_{NO_2}$ or VCD$_{LNO_x}$). In detail, these two AMFs can be calculated as:

$$AMF_{\text{Trop}} = \frac{(1 - f_{effNO_2}) \int_{p_{surf}}^{p_{tp}} w_{clear}(p) NO_2(p)\, dp + f_{effNO_2} \int_{p_{cloud}}^{p_{tp}} w_{cloudy}(p) NO_2(p)\, dp}{\int_{p_{surf}}^{p_{tp}} NO_2(p)\, dp}$$

$$AMF_{\text{LNO}_x} = \frac{(1 - f_{effNO_2}) \int_{p_{surf}}^{p_{tp}} w_{clear}(p) NO_2(p)\, dp + f_{effNO_2} \int_{p_{cloud}}^{p_{tp}} w_{cloudy}(p) NO_2(p)\, dp}{\int_{p_{surf}}^{p_{tp}} LNO_x(p)\, dp}$$

7) Page 7, line 161: "The squall line on 1 September, 2020 was born..."

Sorry for the wrong date. Fixed.

The squall line on **1 September, 2020** was born in the ...

8) Page 10, Fig. 4 caption: "The vertical distributions of the O3 net production rate and tendency..."

Fixed.

The vertical distributions of the O$_3$ **net production rate and tendency** due to ...

9) Page 10, line 198: Regarding "...less significant (< 1%)..." From Table 1, I estimate that for the 2020 case, changes in chemistry affect net O3 production by 0.3% and ≈3% during the life cycle and convective period, resp.

The changes are (16.9-17.1)/17.1 = 1% and (31.9-32.0)/32.0 = 0.3% for the convective period and life cycle, resp (Table R1).

The decreased chemistry contribution is less significant (**≤ 1%**) for the 2020 case which has a smaller lightning density near the station.

10) P10, line 201: "...can certainly enhance the downwind ozone production on the scale of days..."

Fixed.

Note that the LNO$_x$ can certainly enhance the downwind ozone production **on the scale of days** (Pickering et al., 1996; DeCaria et al., 2005).

11) Page 12, Fig. 5 caption: Please state white grid cells are for missing TROPOMI data (no2_scd_flag > 0 ?).

We have modified the caption and added the definition of no2_scd_flag in the Appendix A.

**Fig caption**

... These white grid cells stand for missing TROPOMI data (no2_scd_flag $\neq$ 0), as defined in Appendix A.

**Appendix A: Flag definitions used in this study**

The no2_scd_flag is introduced to make usage of the NO$_2$ slant column (SCD) data easier, by gathering information from a few variables into one flag (Table R3). This flag can thus be used for filtering, though with care as it probably does not cover all possible situation. Here "$\delta$" refers to the SCD error (in mol/m$^2$) and "pqf" stands for processing quality flag.

**Table R3 .** Definition of no2_scd_flag.

| Value | Meaning |
|---|---|
| -1 | no SCD value due to saturation limit exceeded, i.e. pqf=54 |
| 0 | SCD with $\delta < 3.3\times10^{-5}$ & no error reported |
| 1 | SCD with $\delta < 3.3\times10^{-5}$ & error reported: pqf=55 |
| 2 | SCD with $\delta < 3.3\times10^{-5}$ & other error reported, e.g. pqf=41 |
| 3 | SCD with $\delta \geq 3.3\times10^{-5}$ & no error reported |
| 4 | SCD with $\delta \geq 3.3\times10^{-5}$ & error reported: pqf=55 |
| 5 | SCD with $\delta \geq 3.3\times10^{-5}$ & other error reported, e.g. pqf=41 |
| FillValue | no SCD due other error (prior to the Differential Optical Absorption Spectroscopy fit) |

12) Page 13, lines 218-219: "... middle troposphere (MT, 800 hPa to 400 hPa), upper troposphere (UT 400 hPa to 150 hPa)..."

Added the abbreviation of upper troposphere.

The changes in the retrieved AMFs are examined by replacing profiles in three tropospheric layers independently: middle troposphere (MT, 800 hPa to 400 hPa), upper troposphere (**UT**, 400 hPa to 150 hPa), and troposphere (surface to tropopause)

13) Page 13, line 220-221: "...Figure 6 shows that the AMF changes...". Also, Beirle et al. (2009) noted a decrease in sensitivity in the UT due to the NO2/NOx branching ratio.

Added.

Figure 7 shows that the **AMF** changes are mostly controlled by the LNO$_x$ in the UT layer where the detection sensitivity is high **(Beirle et al., 2009; Laughner and Cohen, 2017) and the LNO$_x$ production reaches the peak (Fig. S6)**.

14) Page 13, line 226: "...UT $\Delta$AMFtrop > 20 % exists in Fig. 6bi and biii,"

Fixed.

This explains why UT $\Delta$AMF$_{trop}$ > 20 % exits in Fig. 7b$_i$ and b$_{iii}$,

15) Page 14, Fig 6 caption:
"...is the AMF$_{trop}$ with 500 mol NO per flash relative to 0 mol NO per flash"
"...is the AMF$_{LNO_x}$ with 700 mol NO per flash relative to 500 mol NO per flash"
Also, it would help to see the ovals overplotted on all figures for easier comparison.

Thanks for the suggestion. We have added ovals to the first and third row of Fig. R4, because the $\Delta AMF_{LNO_x}$ is mostly negative and the figure is cleaner than that full of ovals.

[Figure]

**Figure R4 .** The percent differences of AMFs by replacing the a priori $NO_2$ profiles at three layers: middle troposphere (left), upper troposphere (middle), and troposphere (right). $\Delta AMF_{trop}$ is the comparison of the $AMF_{trop}$ with 500 mol NO per flash relative to 0 mol NO per flash. $\Delta AMF_{LNO_x}$ is the comparison of the $AMF_{LNO_x}$ with 700 mol NO per flash relative to 500 mol NO per flash. Three regions are annotated: fresh lightning (blue), downwind of fresh lightning (red), and aged lightning (green). Because of the quite large $AMF_{LNO_x}$ values in pixels with little lightning, $\Delta AMF_{LNO_x}$ is shown over pixels where $0 < AMF_{LNO_x} < 10$.

16) Page 15, Fig 7: Please label the x-axes in (b), (c) and (d).

Added (Fig. R5).

[Figure]

**Figure R5 .** (a) The relationship between cloud pressure and $f_{effNO_2}$ for three regions defined in Fig. 7: fresh lightning region, downwind of fresh lightning, and aged lightning area. (b–d) The a priori $SCD_{tropNO_2}^{LNO_x}/SCD_{tropNO_2}^{noLNO_x}$ and a priori $VCD_{tropNO_2}^{LNO_x}/VCD_{tropNO_2}^{noLNO_x}$ of pixels in these three regions. The $LNO_x$ superscript indicates that the a priori variable is calculated with $LNO_x$ (500 mol NO per flash) and the $noLNO_x$ superscript is without $LNO_x$.

17) Figure S1, caption: "... WRF-Chem simulations for the 2019 and 2020 cases."

Fixed.

Domain and terrain height (m) of the WRF-Chem simulations for the 2019 and 2020 cases.

18) Figure S2: The times/dates in the legend of (a) are not correct

Sorry for the misleading. Because the time resolution of WACCM is 6 hours, the time shown in the legend (Fig. R6) is different from that of ozonesonde.

19) Figure S3, caption: "... in Fig. 2 for 25 July, 2019.""

Fixed (Fig. R7).

[Figure]

**Figure R6 .** (a) Regional mean (118.5°E – 119.5°E, 31.5°N – 32.5°N) preconvection (blue) and postconvection (orange) $O_3$ profiles from the **6-hour** WACCM forecasts. (b) The percent difference of $O_3$ profiles in (a).

[revised manuscript text omitted]

---

## Author Response (AR2)

**Influence of convection on the upper tropospheric $O_3$ and $NO_x$ budget in southeastern China**

**Response to Anonymous Referee #1**

Xin Zhang, Yan Yin, Ronald van der A, Henk Eskes, Jos van Geffen, Yunyao Li, Xiang Kuang, Jeff L. Lapierre, Kui Chen, Zhongxiu Zhen, Jianlin Hu, Chuan He, Jinghua Chen, Rulin Shi, Jun Zhang, Xingrong Ye, and Hao Chen

April 12, 2022

We thank the reviewer for his/her positive comments and very careful reading of our article. The individual corrections suggested are addressed below. The reviewer's comments will be shown in red, our response in blue, and changes made to the paper are shown in **black** block quotes. Unless otherwise indicated, page and line numbers correspond to the original manuscript. Figures, tables, or equations referenced as "R$n$" are numbered within this response; if these are used in the changes to the paper, they will be replaced with the proper number in the revised version.

The authors have satisfactorily answered all my comments and the manuscript has been significantly improved. However, I have still one minor comment:

- I suggested using LIS lightning data to estimate the IC DE of the employed LLS. The auhtors have not done that, but they have alternatively proposed using a constant IC:CG ratio based on literature. I think this is valid. However, they have now written:

"Although CG detection efficiency of ENGLN is not known for this region due to a lack of validation data,"

I do not agree with this phrase because the lightning data from LIS cpuld be used for validation. I suggest the authors remove this phrase before the manuscript is published.

Thanks for the correction and we have modified it. Besides, the color of figures has been updated for persons with colour vision deficiencies.

> IC data will become more accurate if more Chinese total lightning networks, such as Beijing Lightning Network (BLNET; Srivastava et al., 2017), **are available to be compared with lightning imaging sensors (Rudlosky and Shea, 2013; Poelman and Schulz, 2020)**.

**References**

Poelman, D. R. and Schulz, W.: Comparing Lightning Observations of the Ground-Based European Lightning Location System EUCLID and the Space-Based Lightning Imaging Sensor (LIS) on the International Space Station (ISS), Atmos. Meas. Tech., 13, 2965–2977, https://doi.org/10.5194/amt-13-2965-2020, 2020.

Rudlosky, S. D. and Shea, D. T.: Evaluating WWLLN Performance Relative to TRMM/LIS, Geophys. Res. Lett., 40, 2344–2348, https://doi.org/10.1002/grl.50428, 2013.

Srivastava, A., Tian, Y., Qie, X., Wang, D., Sun, Z., Yuan, S., Wang, Y., Chen, Z., Xu, W., Zhang, H., Jiang, R., and Su, D.: Performance Assessment of Beijing Lightning Network (BLNET) and Comparison with Other Lightning Location Networks across Beijing, Atmos. Res., 197, 76–83, https://doi.org/10.1016/j.atmosres.2017.06.026, 2017.